# Competing Bandits in Matching Markets via Super Stability

**Soumya Basu** [1]

## Abstract

We study bandit learning in matching markets with two-sided reward uncertainty, extending prior research primarily focused on single-sided uncertainty. Leveraging the concept of 'super-stability' from Irving (1994), we demonstrate the advantage of the Extended Gale-Shapley (GS) algorithm over the standard GS algorithm in achieving true stable matchings under incomplete information. By employing the Extended GS algorithm, our centralized algorithm attains a logarithmic pessimal stable regret dependent on an instance-dependent admissible gap parameter. This algorithm is further adapted to a decentralized setting with a constant regret increase. Finally, we establish a novel centralized instance-dependent lower bound for binary stable regret, elucidating the roles of the admissible gap and super-stable matching in characterizing the complexity of stable matching with bandit feedback.

## 1. Introduction

The problem of finding a stable matching in a two-sided matching market has gained considerable attention in the past few years (Das & Kamenica, 2005; Liu et al., 2020; Sankararaman et al., 2021; Basu et al., 2021; Liu et al., 2021; Jagadeesan et al., 2021; Hosseini et al., 2024; Ghosh et al., 2024). A major motivation is the two-sided matching market provides a model to study multiple real world systems - such as crowd-sourcing markets like UpWork or Task Rabbit (Li et al., 2019; Zhang et al., 2023), ride-sharing systems (Johari et al., 2021) as well as classical markets like matching students in college programs (Gale & Shapley, 1962), matching organ donors with recipients (Reddy et al., 2013). In a two-sided market there are two types of agents, demand side (users) and supply side (arms). Each agent has an implicit preference order for the other side agents.

However, this preference order is unknown a-priori, and must be elicited from noisy feedback by matching with the other side through repeated interactions. A stable matching is a matching among the users and arms, where there is no *blocking pair* of user $i$ and arm $j$ such that $i$ and $j$ prefer each other compared to their current partner. The objective of the system is to converge to *one* of multiple possible stable matching with low regret.

Unlike most prior works, except a select few (Pagare & Ghosh, 2024; Zhang & Fang), which are limited to user side uncertainty we consider *two-sided uncertainty*, where all users and arms need to learn their respective preferences. Our work also considers general matching markets. In another direction, the existing literature has been limited to combining bandit learning methods with the standard Gale-Shapley algorithm. One major shortcoming of this approach is the need to learn the top $N$ ranks for a general matching market with $N$ users and $K$ arms where $N \leq K$. This does not capture the intrinsic complexity of the underlying stable matching problem. We move beyond the standard Gale-Shapley algorithm, and harness a seminal work of Irving (1994) to construct an algorithm with regret that grows with the intrinsic complexity of the problem.

In order to understand the weakness of Gale-Shapley algorithm let us first review the task at hand. Given the history, at each round users and arms can construct their respective partially recovered rankings, and require to find a stable match using these partial rankings. However, using Gale-Shapley algorithm over a partially recovered ranking, breaking ties in a uncertain manner, the system arrives at a *weakly stable matching*. This is a concept of stability defined in Irving (1994) for partial rankings, where a matching is weakly stable if there is no *blocking pair* of user $i$ and arm $j$ such that $i$ and $j$ *strictly prefers* each other compared to their current partner. The problem with weakly stable matching is as ties in preferences are resolved a weakly stable matching can turn out to have blocking pairs. Therefore, it is not stable under the true full-rankings. The only way to ensure exploration is complete is by resolving the top $N$ ranks for all the agents and ensure that a weakly stable matching is indeed stable under the true full-rankings.

Interestingly, in Irving (1994) we can also find a way to circumvent this issue leveraging the concept of *super stable*

[1]Google, New York. Correspondence to: Soumya Basu <basusoumya@google.com>.

*Proceedings of the 42nd International Conference on Machine Learning*, Vancouver, Canada. PMLR 267, 2025. Copyright 2025 by the author(s).

*matching*. A matching is super stable under partial rankings of the agents if there is no blocking pair of user $i$ and arm $j$ such that $i$ and $j$ *strictly prefers or is indifferent to* each other compared to their current partner. If at any stage a *super stable matching* is found it is ensured that this is a stable matching under the true complete full-rankings. The Extended Gale-Shapley algorithm in Irving (1994) ensures whenever a *super stable matching* exists we can recover one, otherwise determine none exists. This provides us with a way to adaptively exploit when a *super stable matching* is present under the recovered partial rank, or explore more when it is absent.

We first show how Extended Gale-Shapley algorithm can be used with an UCB-LCB based rank recovery in a centralized setting, and prove that we can achieve a logarithmic pessimal stable regret that depends on an instance-dependent *admissible gap* parameter. Notably, our method does not require resolving top $N$ ranks for all agents and adapt to the problem hardness. Next, using a 2-bit shared feedback we show how this centralized algorithm can be modified to a decentralized algorithm at the cost of constant regret increase. Finally, we present a new instance-dependent pessimal stable regret lower bound for general instances in a centralized setting. Our lower bound depends on the *admissible gap*, and shows that *admissible gap* is indeed an intrinsic hardness parameter for bandits in matching markets. It also highlights how the super-stable matching plays a pivotal role in the informational bottlenecks for this problem.

Our main contributions are as follows:

**A new algorithmic pathway:** Gale-Shapley-based bandit algorithms suffer from a key limitation: under partial preference information, they can only guarantee convergence to a weakly stable matching, which may not be a true stable matching. However, as shown by Irving (1994), given a partial ranking, it is possible to find a super-stable matching if one exists or to determine that none exists. Crucially, if a super-stable matching exists for a partial rank that is compatible with the true full rank, then that super-stable matching is guaranteed to be a true stable matching. Since a UCB-LCB-based partial rank is compatible with the true full rank with high probability, the resulting super-stable matching (if found) incurs no pessimal stable regret. This insight opens a *novel algorithmic pathway* for addressing bandit problems in two-sided matching markets with uncertainty.

**Algorithms for two sided uncertainty:** Building on this insight, we first develop a centralized algorithm that constructs a UCB-LCB-based partial rank and attempts to recover a super-stable matching using the Extended-GS algorithm (Irving, 1994). If no such super-stable matching exists, the algorithm defaults to round-robin exploration. Exploiting the structure of the super-stable matching set due

to Spieker (1995), we define the set of admissible partial ranks—partial ranks that are compatible with the true full rank, and whose compatible full ranks all share at least one stable matching with the true full rank. If the UCB-LCB-based partial rank falls within this admissible set, a true stable matching is guaranteed. Let $\Delta_{\mathcal{A}}$ represent the maximum of the minimum gaps, where the minimum gap for a given partial rank is the smallest difference between any two unequally ranked arms for any user (and vice versa). We demonstrate that each user and arm incurs an expected cumulative pessimal stable regret of $O(K \log(T)/\Delta_{\mathcal{A}}^2)$ over $T$ rounds. This improves upon existing results that achieve $O(K \log(T)/\Delta_{\min}^2)$, as $\Delta_{\mathcal{A}} \geq \Delta_{\min}$. Our experiments confirm the superior performance of our proposed algorithm compared to existing Gale-Shapley-based approaches. We then demonstrate how, using only two shared boolean flags, users and arms can emulate the centralized algorithm in a decentralized setting, incurring only an additional $O(N^2)$ cumulative regret, independent of the time window. We also discuss the communication trade-offs involved. This adaptation from centralized to decentralized settings may have broader applicability in two-sided matching markets.

**Instance-dependent regret lower bound:** We establish the first instance-dependent regret lower bound for stable matching in the centralized (and consequently decentralized) setting under two-sided uncertainty. This bound focuses on binary stable regret, quantifying the number of times a stable matching is not achieved. Building upon the framework of Combes et al. (2017) we formulate this lower bound as an optimization problem over possible matching market instances, introducing a novel constraint set. Crucially, instances sharing a stable matching with the true instance do not contribute to the lower bound. We derive explicit lower bounds for specific market structures, including general serial dictatorships and markets with redundant arms, demonstrating a scaling of $\Omega(K_{eff} \log(T)/\Delta_{eff}^2)$, where $K_{eff}, \Delta_{eff}$ are instance-dependent parameters. Through a dual formulation, we reveal a fundamental connection between our lower bound and specific set covers of the boundary of the *admissible partial rank* set. While a tight regret bound remains open, we anticipate that the structure of admissible partial rank set will be pivotal in achieving it.

## 2. Problem Formulation

We consider the matching markets with two-sided uncertainty. There are $N$ users and $K$ arms. We are interested in a setting where the number of users is less or equal to the number of arms $N \leq K$.

Each user $i \in [N]$ has a valuation for an arm $j \in [K]$ denoted by $\mu_{i,j}$. Similarly, each arm $j \in [K]$ has a valuation for a user $i \in [N]$ denoted by $\gamma_{j,i}$. These valuations are a

priori unknown to the users and arms, and can be learned only by matching with the other side. We denote by $F_{u,i}$ as the preference full-rank of user $i \in [N]$ over the arms $[K]$, and $F_{a,j}$ as the preference full-rank of arm $j \in [K]$ over the users $[N]$. We denote the set of full-ranks by $(F_u, F_a) = (\{F_{u,i} : i \in [N]\}, \{F_{a,j} : j \in [K]\})$.

The system evolves in rounds, while in each round a user can match with at most one arm, and vice versa. When an arm is matched with multiple users in a round, there is a collision, and all the users involved in the collision receive zero reward, and a collision signal. Let us call the match for user $i \in [N]$ in round $t$ as $m_i(t) \in [K] \cup \{\emptyset\}$, and the match for an arm $j \in [K]$ in round $t$ as $m_j^{-1}(t) \in [N] \cup \{\emptyset\}$. Here, $m_i(t) = \emptyset$ implies the user $i$ is unmatched, and $m_j^{-1}(t) = \emptyset$ implies arm $j$ remains unmatched. For each pair $(i, j) \in m(t)$ for $i \in [N]$ and $j \in [K]$ the user $i$ and arm $j$ receives noisy rewards $Y_i(t)$ and $\tilde{Y}_j(t)$, respectively. The rewards are given as

$$Y_i(t) = \mu_{i,m_i(t)} + \eta_{i,m_i(t)}(t) \text{ if } m_i(t) \neq \emptyset, \text{ else } 0,$$
$$\tilde{Y}_j(t) = \gamma_{j,m_j^{-1}(t)} + \eta'_{j,m_j^{-1}(t)}(t) \text{ if } m_j^{-1}(t) \neq \emptyset, \text{ else } 0,$$

where $\eta_{i,j}(t)$ and $\eta'_{j,i}(t)$ are independent 1-subgaussian noise for $i \in [N]$ and $j \in [K]$.

We consider two versions of the system that differs in how the matching is formed in each round.

**Centralized:** In this setting, in each round the arms and users reveal their learned preferences to a central platform. The central platform arrives to a matching in that round.

**Decentralized:** In this setting, in each round the users can propose to the arms (possibly multiple). Each arm can accept and match with at most one user, while rejecting all the unmatched proposing users. Additionally, users have access to 2 shared flags (binary semaphores)[1]. After receiving the signals from the arms, the users modify the shared bits and finally use the updated shared bits to match with arms.

In each round, the objective of all the users and the arms are trying to construct a stable match defined as below.

**Definition 2.1** (Stable Match). A matching $M$ is called stable under a full rank $(F_u, F_a)$ if there is no pair user $i \in [N]$, and arm $j \in [K]$ such that simultaneously user $i$ prefers to arm $j$ over her match $M_i$, and arm $j$ prefers user $i$ over its match $M_j^{-1}$. The set of all super-stable matching of the partial rank $(F_u, F_a)$ is denoted as $\text{Stable}(F_u, F_a)$.

In general, there are multiple stable matching given a full rank $(F_u, F_a)$ which we denote by $Stable(F_u, F_a)$. There

---

[1] Shared information is used in the literature to avoid technical difficulties pertaining communication design, e.g. (user, arm) broadcast in (Liu et al., 2021; Kong & Li, 2023; Pagare & Ghosh, 2024). It is possible to make the algorithm fully decentralized with an additional $O(\log(T))$ regret in $T$ rounds.

exists a user-pessimal matching $M(u) \in Stable(F_u, F_a)$ such that for each user $i \in [N]$ the reward $\mu_{i,pess} = \mu_{i,M_i(u)}$ is the lowest among all possible partner in a stable matching. Similarly, we define the stable arm-pessimal match $M(a)$ and the arm-pessimal reward $\gamma_{j,pess} = \gamma_{j,M_j^{-1}(a)}$ for each arm $j \in [K]$. We define the expected pessimal stable regret for the users $i \in [N]$ and the arms $j \in [K]$ in $T$ rounds as

$$\mathbb{E}[R_{u,i}(T)] = T\mu_{i,pess} - \sum_{t=1}^{T} \sum_{j \in [K]} \mu_{i,j} \mathbb{P}(m_i(t) = j),$$

$$\mathbb{E}[R_{a,j}(T)] = T\gamma_{j,pess} - \sum_{t=1}^{T} \sum_{i \in [N]} \gamma_{j,i} \mathbb{P}(m_j^{-1}(t) = i).$$

We also consider the *binary stable regret* where regret of failure to find a stable match is counted as 1, and otherwise 0 each round, i.e.

$$\mathbb{E}[R_{0/1}(T)] = \sum_{t=1}^{T} \mathbb{P}(m(t) \notin \text{Stable}(F_u, F_a)).$$

### 2.1. Partial Rank and Super Stable Matching

In this section, we present the structure of the super-stable matching due to Spieker (1995). In order to state the results we first define partial rank and super-stable matching.

**Definition 2.2** (Partial Rank). A partial rank $(P_u, P_a)$ over $N$ users and $K$ arms is defined as a set of directed acyclic graphs (DAG) $P_{u,i}$ for each $i \in [N]$ and $P_{a,j}$ for each $j \in [K]$. Each DAG $P_{u,i}$ is defined by a set of directed edges $(j, j') \subset [K] \times [K]$, and satisfies $j >_{P_{u,i}} j'$ *if and only if* there exists a directed path from $j$ to $j'$. Each DAG $P_{a,j}$ is defined analogously. Let us define the set of all partial ranks for $N$ users, and $K$ arms as $\mathcal{P}(N, K)$.

**Definition 2.3** (Compatible Full Rank). A partial rank $(P'_u, P'_a)$ is *compatible* with a partial rank $(P_u, P_a)$ if for any pair of users $i, i' \in [N]$, and any pair of arms $j, j' \in [K]$, $j \underset{P'_{u,i}}{>} j' \implies j' \underset{P_{u,i}}{\not>} j$, and $i \underset{P'_{a,j}}{>} i' \implies i' \underset{P_{a,j}}{\not>} i$.
The set of all full ranks *compatible* with a partial rank $(P_u, P_a)$ is denoted as $\text{FullRank}(P_u, P_a)$.

If a full rank $(F_u, F_a)$ is *compatible* with a partial rank $(P_u, P_a)$, then there exists a way of breaking ties (by adding more directed edges) in the partial rank $(P_u, P_a)$ to reach $(F_u, F_a)$. The reversal of the process takes us from a full rank to a compatible partial rank.

We now define the notion of super-stability for a partial rank, and follow it up with a proposition due to Spieker (1995) that shows how it relates to the compatible full ranks.

**Definition 2.4** (Super-Stable Match). A matching $M$ is called super-stable under a partial rank $(P_u, P_a)$ if there is

no pair user $i \in [N]$, and arm $j \in [K]$ such that simultaneously user $i$ prefers *or is indifferent to* arm $j$ over her match $M(i)$, and arm $j$ prefers *or is indifferent to* user $i$ over its match $M^{-1}(j)$. The set of all super-stable matching of the partial rank $(P_u, P_a)$ is denoted as SuperStable$(P_u, P_a)$.

**Proposition 2.5.** *[Adapted from Spieker (1995)] For a partial ranking $(P_u, P_a)$, the set of super-stable matching is the (possibly empty) intersection of all the stable matching of full ranks $(F_u, F_a)$ compatible with $(P_u, P_a)$,*

$$\text{SuperStable}(P_u, P_a) \\ = \bigcap_{(F_u, F_a) \in \text{FullRank}(P_u, P_a)} \text{Stable}(F_u, F_a).$$

From the above proposition, given a full rank we define a set of partial match, namely *admissible partial rank*, for which a super-stable match is always stable under the specific full rank.

**Definition 2.6** (Admissible Partial Rank). For a given full rank $(F_u, F_a)$, we define a partial rank $(P_u, P_a)$ to be *admissible* if and only if $(F_u, F_a) \in$ FullRank$(P_u, P_a)$ and SuperStable$(P_u, P_a) \neq \emptyset$. We define the set of admissible partial rank instances as $\mathcal{A}(F_u, F_a)$.

It follows from Proposition 2.5 that for any partial ranking in the set $\mathcal{A}(F_u, F_a)$ a super-stable match is a stable match for the true Stable matching instance $(F_u, F_a)$.

**Corollary 2.7.** *For a full rank $(F_u, F_a)$, for any admissible partial rank $(P_u, P_a) \in \mathcal{A}(F_u, F_a)$ and each super-stable matching $M \in$ SuperStable$(P_u, P_a)$ we have $M \in$ Stable$(F_u, F_a)$.*

## 3. Centralized Two-Sided Matching Bandit

In this section, we present a centralized algorithm that recovers a stable match with uncertainty of the rewards on both sides of the market. Each agent and arm maintain their own partial rankings based on UCB-LCB (similar to Kong & Li (2023)). Similarly to Liu et al. (2020) in each round, users share their UCB-LCB-based partial rankings with the centralized platform. The centralized platform then uses the partial ranking and finds a super-stable match if it exists using the Extended-Gale Shapley algorithm in Irving (1994) (given as Algorithm 2). If such a match does not exist, then using a round-robin schedule, the centralized platform explores.

We update the UCB, LCB, and partial rank estimates as follows for all $i \in [N]$ and all $j \in [K]$:

$$\mu\text{-}ucb_{i,j}(t) = \hat{\mu}_{i,j}(t) + \sqrt{6\log(t)/n_{i,j}(t)},$$
$$\mu\text{-}lcb_{i,j}(t) = \hat{\mu}_{i,j}(t) - \sqrt{6\log(t)/n_{i,j}(t)} \qquad (1)$$
$$\gamma\text{-}ucb_{j,i}(t) = \hat{\gamma}_{j,i}(t) + \sqrt{6\log(t)/n_{i,j}(t)},$$

---

**Algorithm 1:** Centralized Two-Sided Matching Bandit

1 **Exploration Index:** $\tau_{ex} \leftarrow 0$
2 **Initial Ranking:** $P_{u,i}(1) \leftarrow [[K]]$ for all $i \in [N]$, $P_{a,i}(1) \leftarrow [[N]]$ for all $i \in [K]$
3 **for** $t \geq 1$ **do**
4    The centralized platform receives the partial ranks:
5      users, $\mathcal{P}_u(t) = \{P_{u,i}(t) : i \in [N]\}$ and
6      arms $\mathcal{P}_a(t) = \{P_{a,j}(t) : j \in [K]\}$.
7    Obtain $M_{\text{stable}} \leftarrow$ EXTENDED-GS$(\mathcal{P}_u(t), \mathcal{P}_a(t))$
8    **if** $M_{\text{stable}} = \emptyset$ **then**
9      Play: $M(t) \leftarrow \{m_i(t) = (i + \tau_{ex}) \mod K\}$
10      Increase exploration index: $\tau_{ex} \leftarrow \tau_{ex} + 1$
11    **else**
12      Trim and play matching:
       $M(t) \leftarrow \{(i, \arg\max_{j \in m_i} \mu\text{-}ucb_{i,j}(t)) : (i, m_i) \in M_{\text{stable}}\}$
13    **for** *each pair* $(i, j) \in M(t)$ **do**
14      User $i$ receives reward $Y_{i,j}(t)$, and arm $j$ receives reward $\tilde{Y}_{j,i}(t)$
15      User $i$ and arm $j$ updates the partial rank $P_{u,i}(t+1)$ and $P_{a,j}(t+1)$ (Eq. (3))

---

$$\gamma\text{-}lcb_{j,i}(t) = \hat{\gamma}_{j,i}(t) - \sqrt{6\log(t)/n_{i,j}(t)} \qquad (2)$$

$$P_{u,i}(t) = \begin{cases} j' > j \text{ if } \mu\text{-}ucb_{i,j}(t) < \mu\text{-}lcb_{i,j'}(t), \\ j' < j \text{ if } \mu\text{-}ucb_{i,j'}(t) < \mu\text{-}lcb_{i,j}(t), \\ j' = j \text{ otherwise.} \end{cases}$$

$$P_{a,j}(t) = \begin{cases} i' > i \text{ if } \gamma\text{-}ucb_{j,i}(t) < \gamma\text{-}lcb_{j,i'}(t), \\ i' < i \text{ if } \gamma\text{-}ucb_{j,i'}(t) < \gamma\text{-}lcb_{j,i}(t), \qquad (3) \\ i' = i \text{ otherwise.} \end{cases}$$

Here for round $t$, for $i \in [N]$, arm $j \in [K]$, for user $i$ the mean reward for arm $j$ is $\hat{\mu}_{i,j}(t)$, for arm $j$ the mean reward for user $i$ is $\hat{\gamma}_{j,i}(t)$, and number of matches between user $i$ and arm $j$ is $n_{i,j}(t)$ as defined in the Appendix C.

### 3.1. Regret Upper Bound

We obtain a logarithmic regret with two-sided uncertainty. Our result relies on a few key properties of the system. Firstly, we know from Kong & Li (2023) that with high probability the LCB and UCB-based partial ranking contains the true ranking for each of the user and arm. Next, by the seminal work of Irving (1994), the Extended Gale-Shapley algorithm can retrieve a super-stable matching if there exists one under a given partial ranking, or declare if there is no such super-stable matching. Finally, the key observation is that a super-stable matching for a partial ranking is always a stable (not necessarily the user-optimal) matching for any full ranking contained by the specific partial ranking (c.f.

Spieker (1995)). Our exploration is forced when there is no super-stable matching available.

We now define the gaps in our system that will appear in our regret upper bounds. Note that all the gaps we mention is with respect to the underlying rewards $(\boldsymbol{\mu}, \boldsymbol{\gamma})$.

**Definition 3.1** (Minimum Gap). For a system with rewards $(\boldsymbol{\mu}, \boldsymbol{\gamma})$ and a partial rank $(P_u, P_a)$, the minimum gap $\Delta_{\min}(P_u, P_a; \boldsymbol{\mu}, \boldsymbol{\gamma})$ is defined as the minimum gaps among the users and arms with different ranks, i.e.,

$$\Delta_{\min}(P_u, P_a; \boldsymbol{\mu}, \boldsymbol{\gamma})$$
$$= \min \Big( \min_{\substack{i \in [N], \\ j \neq j' \text{ in } P_{u,i}}} |\mu_{i,j} - \mu_{i,j'}|, \min_{\substack{j \in [K], \\ i \neq i' \text{ in } P_{a,j}}} |\gamma_{j,i} - \gamma_{j,i'}| \Big).$$

**Definition 3.2** (Admissible Gap). Consider a system with rewards $(\boldsymbol{\mu}, \boldsymbol{\gamma})$ and corresponding full rank $(F_u, F_a)$. The admissible gap $\Delta_{\mathcal{A}}(\boldsymbol{\mu}, \boldsymbol{\gamma})$ is defined as the largest minimum gap for the admissible partial rankings of $(F_u, F_a)$,
$$\Delta_{\mathcal{A}}(\boldsymbol{\mu}, \boldsymbol{\gamma}) = \max_{(P_u, P_a) \in \mathcal{A}(F_u, F_a)} \Delta_{\min}(P_u, P_a; \boldsymbol{\mu}, \boldsymbol{\gamma}).$$

We also need to define the width of the rewards of users or arms that have the same partial order.

**Definition 3.3** (Overlap Width). For a system with rewards $(\boldsymbol{\mu}, \boldsymbol{\gamma})$ and a partial rank $(P_u, P_a)$, overlap width $\mathcal{W}_{ov}(P_u, P_a; \boldsymbol{\mu}, \boldsymbol{\gamma})$ as the maximum variation of rewards among overlapping users and arms with an equivalent partial rank, i.e.,

$$\mathcal{W}_{ov}(P_u, P_a; \boldsymbol{\mu}, \boldsymbol{\gamma})$$
$$= \max \Big( \max_{\substack{i \in [N], \\ j = j' \text{ in } P_{u,i}}} |\mu_{i,j} - \mu_{i,j'}|, \max_{\substack{j \in [K], \\ i = i' \text{ in } P_{a,j}}} |\gamma_{j,i} - \gamma_{j,i'}| \Big).$$

We next show that the set $\mathcal{A}(F_u, F_a)$ has the following 'completeness' property with respect to reward gap.

**Lemma 3.4.** *Consider the system with rewards $(\boldsymbol{\mu}, \boldsymbol{\gamma})$ and the corresponding full rank $(F_u, F_a)$. For a partial rank $(P_u, P_a)$ if $(F_u, F_a) \in \text{FullRank}(P_u, P_a)$ and the overlap width $\mathcal{W}_{ov}(P_u, P_a; \boldsymbol{\mu}, \boldsymbol{\gamma}) < \Delta_{\mathcal{A}}(\boldsymbol{\mu}, \boldsymbol{\gamma})$ then $(P_u, P_a) \in \mathcal{A}(F_u, F_a)$.*

The regret upper bound proof shows that once we have enough exploration, such that we can resolve a gap of $\Theta(\Delta_{\mathcal{A}}(\boldsymbol{\mu}, \boldsymbol{\gamma}))$ with high probability in round $t$, the partial rank $(\mathcal{P}_u(t), \mathcal{P}_a(t))$ lies in the admissible set $\mathcal{A}(F_u^*, F_a^*)$. Hence, the Algorithm 1 always lands on a super-stable matching, as the set $\text{SuperStable}(\mathcal{P}_u(t), \mathcal{P}_a(t)) \neq \emptyset$ as per Corollary 2.7. Furthermore, due to Corollary 2.7 we know that $\text{SuperStable}(\mathcal{P}_u(t), \mathcal{P}_a(t)) \subseteq \text{Stable}(F_u^*, F_a^*)$. Therefore, the algorithm suffers no stable regret. Finally, once the number of exploration satisfies $N_{explore}(t) > K + \frac{96K \log(T)}{\Delta_{\mathcal{A}}^2(\boldsymbol{\mu}, \boldsymbol{\gamma})}$ we show that $(\mathcal{P}_u(t), \mathcal{P}_a(t)) \in \mathcal{A}(F_u^*, F_a^*)$, i.e. the recovered partial rank lies in the set of admissible

partial rank instances of the true rankings. This culminates in the following regret bound.

**Theorem 3.5** (Centralized Regret Bound). *The cumulative regret of Algorithm 1 in $T$ rounds for the true stable matching instance with no ties $(F_u^*, F_a^*)$ satisfies the following upper bound.*

$$\max_{i \in [N], j \in [K]} \left( \frac{\mathbb{E}[R_{u,i}(T)]}{\mu_{i, pess} - \mu_{i, \min}}, \frac{\mathbb{E}[R_{a,j}(T)]}{\gamma_{j, pess} - \gamma_{j, \min}} \right) \leq \mathbb{E}[R_{0/1}(T)],$$

$$\mathbb{E}[R_{0/1}(T)] \leq \left( K + \frac{NK\pi^2}{3} + \frac{96K \log(T)}{\Delta_{\mathcal{A}}^2(\boldsymbol{\mu}, \boldsymbol{\gamma})} \right).$$

**Improved Regret Bound:** We first note that for the partial rank where the top user for each arm, and the top arms for each user are separated we always have the user-optimal matching as a super stable-matching. Hence, $\Delta_{\min} \leq \Delta_{\mathcal{A}}$ holds for all the instances. For general instances, it is not possible to improve this relationship, as there are instances where they are equal. We now present a motivating example that shows stark separation between these two quantities.

Consider an example with $N$ users and $N$ arms, for some $N \geq 3$. Fix an $\varepsilon \in (0, 1/2)$. For each user $i \in [N]$ let $i$ and $(i + 1) \mod N$ be the top 2 arms with the respective rewards $\mu_{i,i} = (1 - \varepsilon)$ and $\mu_{i,(i+1) \mod N} = (1 - 2\varepsilon)$. The remaining arms $j \in [N] \setminus \{i, (i + 1) \mod N\}$ all have mean reward $\mu_{i,j} = \varepsilon$. For each arm $j \in [N]$, let $\gamma_{j,(j-1) \mod N} = \varepsilon$ and $\gamma_{j,i} = (1 - \varepsilon)$ for all users $i \neq (j - 1) \mod N$. For this instance we have $\Delta_{\min} = \varepsilon$. One stable matching for this instance is given as $\{(i, i) : i \in [N]\}$.

Next, let us consider the partial ranks for each user $i$,

$$P_{u,i} = \{ i > j, (i + 1) \mod N > j :$$
$$\forall j \in [N] \setminus \{i, (i + 1) \mod N\} \},$$

and the partial ranks for each arm $j$

$$P_{a,j} = \{ i > (j - 1) \mod N : \forall i \in [N] \setminus \{(j - 1) \mod N\} \}.$$

For the above partial ranks, we have $\Delta_{\min}(P_u, P_a, \boldsymbol{\mu}, \boldsymbol{\gamma}) = (1 - 2\varepsilon)$. Moreover, this partial rank $(P_u, P_a)$ has $\{(i, i) : i \in [N]\}$ as a super-stable matching. Therefore, $(P_u, P_a)$ is admissible, and the gap $\Delta_{\mathcal{A}} \geq (1 - 2\varepsilon)$. The ratio $\frac{\Delta_{\mathcal{A}}}{\Delta_{\min}} = \frac{1}{\varepsilon} - 2$ is unbounded as $\varepsilon \to 0$. This shows that there can be an arbitrary separation between $\Delta_{\min}$ and $\Delta_{\mathcal{A}}$.

## 4. Decentralized Two-Sided Matching Bandit

In this section, we convert the centralized algorithm to a decentralized version where the users use two global binary flags *restart* and *success* to coordinate. We present Algorithm 3 used by a user $i$, and Algorithm 4 used by an arm $j$ in the Appendix C due to the lack of space.

Our algorithm proceeds in phases (of at most $N^2$ steps each) and simulates the Algorithm 2 in each phase in a staggered

manner. An ongoing phase is terminated by setting the *restart* flag to True by a user, and as the flag *restart* is shared the change of phase is always in sync across arms and users. Each phase $\tau_\ell$, starting at time $t + 1$, begins with setting the initial partial rank of each user $i$ to $P_{u,i}(t)$ and each arm $j$ to $P_{a,j}(t)$ (the UCB-LCB based partial ranks obtained at the end of previous phase). This is similar to the centralized algorithm starting a round with updated partial ranks.

Next, a user $i$ reads the flag *success* which if False indicates the phase $\tau_\ell$ is used for exploration. Otherwise, the *success* flag being True indicates the previous phase ended with a super-stable match. This implies at least one arm accepted proposal of user $i$, and the user chooses as the arm with highest UCB index among accepting arms as it's match $M_i(\tau_\ell)$ for the phase $\tau_\ell$.

Inside the phase $\tau_\ell$ in each round, we have the following alternating steps between users and arms.
$(i)$ **A user** $i$ first propose to the 'source' nodes of the (updated) partial rank $PR_{u,i}(\tau_\ell)$ (line 14 Alg. 3).
$(ii)$ **An arm** $j$ receives the proposals and computes the 'source' nodes (i.e., the non-dominated nodes) $I^*(j)$ among the proposing nodes $\tilde{S}_j(t)$ and (updated) partial rank $PR_{a,j}(\tau_\ell)$ (line 10 Alg. 4). If there is a unique 'source' node $i^*(j)$, that node is accepted by arm $j$ and all nodes dominated by $i^*(j)$ are deleted from $PR_{a,j}(\tau_\ell)$ (line 11-14 Alg. 4). Otherwise, with multiple proposals all proposing users are rejected, and the 'tail' nodes in partial rank $PR_{a,j}(\tau_\ell)$ are deleted (line 15-17 Alg. 4).
$(iii)$ **A user** $i$ receives the accept or reject signals from the proposed arms, and deletes rejecting arms from $PR_{u,i}(\tau_\ell)$ (line 15-16 Alg. 3). Next, it determines whether to explore or exploit based on $explore(\tau_\ell)$. If it explores then matches with the arm $m_i(t) = (i + \tau_{ex}) \mod K$, otherwise matches with $m_i(t) = M_i(\tau_\ell)$ (line 17-20 Alg. 3). We note that $\tau_{ex}$ is also in sync for each user as $explore(\tau_\ell)$ and phases are in sync. Next, the user and the arms, if matched, observe their respective rewards, and updates the partial ranks $P_{u,i}(t)$ for users $i \in [N]$, and $P_{a,j}(t)$ for arms $j \in [K]$. (line 22-23 Alg. 3, and line 21-23 Alg. 4)

In the *global communication* phase the users play an active role. First, if a user $i$ is rejected by all proposed arms then she sets *success* to False.[2] Next, the *success* flag is read by user $i$. If *success* is True (each user has a prospective match) or if $PR_{u,i}(\tau_\ell)$ is empty the phase terminates. User $i$ sets *restart* to True.

Finally, the *restart* flag is read, by each user and arm. If the flag is True the system enters a new phase. Otherwise, the old phase continues.[3]

---

[2]We require users not changing the flags to also update the flags, so that shared flags are ready to be read.

[3]To trigger a system-wide 'restart' in fully decentralized man-

Our regret upper bound for the decentralized system follows the centralized system closely. We argue that each phase mimics one round of the centralized system. After $O\big(K\log(T)/\Delta_{\mathcal{A}}^2(\boldsymbol{\mu}, \boldsymbol{\gamma})\big)$ many rounds of exploration, any phase with high probability ends in finding a true stable matching, similar to Lemma E.4. The final regret bound in the decentralized case is given as follows.

**Theorem 4.1** (Decentralized Regret Bound). *For a true stable matching instance $(F_u^*, F_a^*)$ when the users follow Algorithm 3, and the arms follow Algorithm 4, the cumulative regret in $T$ rounds satisfies the following upper bound.*

$$\max_{i \in [N], j \in [K]} \left( \frac{\mathbb{E}[R_{u,i}(T)]}{\mu_{i,pess} - \mu_{i,\min}}, \frac{\mathbb{E}[R_{a,j}(T)]}{\gamma_{j,pess} - \gamma_{j,\min}} \right) \leq \mathbb{E}[R_{0/1}(T)],$$

$$\mathbb{E}[R_{0/1}(T)] \leq \left( 1 + N^2 + K + \frac{NK\pi^2}{3} + \frac{96K\log(T)}{\Delta_{\mathcal{A}}^2(\boldsymbol{\mu}, \boldsymbol{\gamma})} \right).$$

**Balancing Communication, and Regret:** We rely on a 2-bit communication protocol per round in order to mimic the centralized system. However, it is possible to make the communication sparse, by forcing the communication to happen once every phase, and force the phases to have arbitrary lengths. The idea is to perform the Global communication if round $t$ is multiple of a pre-determined period, say $L$. This will increase the regret constant from $(1 + N^2)$ to $(1 + L)$. However, this requires working with older matching $M_i(\tau_\ell)$ for a user $i$, which can be non-optimal from user $i$'s point of view. Thus it can create some tension between communication and incentives. Exploring this tension is left as an interesing avenue of future work.

**Experimental Validation:** We numerically study the behavior of the proposed centralized algorithm. Due to lack of space we defer the details of our experiment in Appendix B. As the decentralized algorithm is guaranteed to have only a regret $O(N^2)$ away from the centralized one, we omit the decentralized algorithm. We compare against centralized Explore-then-Gale Shapley (ETGS) algorithm for fair comparison.[4] Extended-GS significantly outperforms ETGS by quickly identifying a viable partial ranking during the exploration phase. This leads to lower regret than ETGS, which needs a full ranking of the top N items before achieving comparable performance.

**Regret Comparison:** We end this section by comparing the regret guarantees of a selected few works in this domain in Table 1. We provide the first regret upper bound that

---

ner, a user broadcasts a RESTART signal to all arms in a single round. Upon receipt, an arm echoes the RESTART signal to all proposing users. Consequently, within one additional round, all users receive the RESTART signal, achieving a fully-decentralized 'restart' flag setting. A congruent strategy can be adopted by users for managing the 'success' flag.

[4]A decentralized version of this algorithm with shared global flags (aka blackboard) is presented in Algorithm 2 in (Pagare & Ghosh, 2024).

works for two sided general matching markets and depends on the instance dependent intrinsic gap $\Delta_{\mathcal{A}}$. Note that our algorithm uses the 2-bit feedback which is more restrictive than the (user, arm) broadcast where all the users and arms can observe the matching that occurs in each round. We also provide the first centralized regret lower bound that depends on gaps (which we call $\Delta_{\mathcal{A},avg}$ for simplicity) related to $\Delta_{\mathcal{A}}$. See Section 5 for details.

# 5. Regret Lower Bound

We develop an instance-dependent lower bound for *binary stable regret* for the centralized setting. A pessimal stable regret lower bound in general instances is not meaningful for the learning task of finding the stable matching. For example, there may exists a set of matchings which are not stable but can be scheduled to achieve negative pessimal stable regret. Hence, *binary stable regret* which is always positive is the right quantity to lower bound for the bandit learning problem in matching markets.

We adapt the framework in Combes et al. (2017) to our multi-agent setup. The detailed formulation is provided in the Appendix F. Our system is parameterized by $\theta = (\boldsymbol{\mu}, \boldsymbol{\gamma}) \in \Theta := [0,1]^{2NK}$. We focus on Bernoulli rewards with appropriate means for ease of exposition. For any $(i,j) \in [N] \times [K]$, the term $kl(\theta, \lambda; (i,j))$ denotes the sum of KL-divergence between the $\theta$ and $\lambda$ instances for the rewards associated with the match $(i,j)$ and satisfies $kl(\theta, \lambda; (i,j)) = kl(\mu_{i,j}(\theta), \mu_{i,j}(\lambda)) + kl(\gamma_{j,i}(\theta), \gamma_{j,i}(\lambda))$. This can be extended to the $kl$ divergence given a matching $\mathcal{M}$ as $kl(\theta, \lambda; \mathcal{M}) = \sum_{(i,j) \in \mathcal{M}} kl(\theta, \lambda; (i,j))$.

We need to take extra care while adapting the main results in Combes et al. (2015) as we are dealing with multiple solutions. To that end, we note that Graves & Lai (1997), from which Combes et al. (2015) is adapted, allows for switching between optimal stationary policies without any cost. Moreover, while defining 'bad' parameter space Graves & Lai (1997) considers the parameter that does not share an optimal solution with the true parameter, $\theta$, and which has no 'divergence' with $\theta$ while playing any one of the optimal policy. Therefore, our 'bad' parameter space for a given $\theta$ becomes

$$\Lambda(\theta) = \{\lambda \in \Theta : \underbrace{kl(\theta, \lambda; \mathcal{M}) = 0, \forall \mathcal{M} \in \text{Stable}(\theta)}_{\text{for all stable match of } \theta \text{ divergence is } 0};$$
$$\underbrace{\text{Stable}(\lambda) \cap \text{Stable}(\theta) = \emptyset}_{\text{no common solution}}\}.$$

We now want to formulate the set $\Lambda(\theta)$ using the admissible sets. If $\text{Stable}(\lambda) \cap \text{Stable}(\theta) = \emptyset$, that means any partial rank $(P_u, P_a)$ such that both $\lambda$ and $\theta$ are compatible to $(P_u, P_a)$[5] has no super-stable match. We note that for any

---

[5] An instance $\theta$ is compatible to a partial rank means the un-

$\lambda'$ such that $\text{Stable}(\lambda') \cap \text{Stable}(\theta) \neq \emptyset$ we can create a partial rank $(P'_u, P'_a)$ (by keeping only the shared pairwise inequalities of $\theta$ and $\lambda'$) which lies in $\mathcal{A}(\theta)$. Hence,

$$\cup_{(P_u, P_a) \in \mathcal{A}(\theta)} \text{FullRank}(P_u, P_a) = $$
$$\{\lambda : \text{Stable}(\lambda) \cap \text{Stable}(\theta) \neq \emptyset\}.$$

It follows that the *admissible set is regret free*. As there can be no divergence for any matched pairs in a true stable matching that narrows the 'bad' set of parameters further. We first define the 'Locked' edges as follows.

**Definition 5.1.** For any instance $\theta \in \Theta$, we define $\text{Locked}(\theta)$ as the set of all user-arm pairs $(i,j)$, such that $(i,j) \in \mathcal{M}$ for some match $\mathcal{M} \in Stable(\theta)$.

The set $\Lambda(\theta)$ can be equivalently stated as

$$\Lambda(\theta) = \Big\{\lambda \in \Theta : \lambda \notin \underset{(P_u, P_a) \in \mathcal{A}(\theta)}{\cup} \text{FullRank}(P_u, P_a); \forall (i,j)$$
$$\in \text{Locked}(\theta) : \mu_{ij}(\theta) = \mu_{ij}(\lambda), \gamma_{ji}(\theta) = \gamma_{ji}(\lambda)\Big\}. \quad (4)$$

A policy $\pi$ is *uniformly good* if for any $\theta = (\boldsymbol{\mu}(\theta), \boldsymbol{\gamma}(\theta))$ the algorithm $\pi$ has $O(T^\alpha)$ regret for any $\alpha > 0$. We now state our regret lower bound below for any uniformly good policy.

**Theorem 5.2** (Centralized Regret Lower Bound). *For any instance $\theta \in \Theta$, for any uniformly good policy $\pi$ the binary stable regret is lower bounded as $\liminf_{T \to \infty} \frac{\mathbb{E}[R^\pi_{0/1}(T; \theta)]}{\log(T)} \geq c(\theta)$, where $c(\theta)$ minimizes the following optimization problem*

$$\min_{\eta(M) \geq 0} \sum_{M \in \text{Match}(N,K) \setminus \text{Stable}(\theta)} \eta(M), \text{ s.t.}$$
$$\sum_{(i,j) \in [N] \times [K]} \sum_{M \ni (i,j)} \eta(M) kl(\theta, \lambda; (i,j)) \geq 1, \forall \lambda \in \Lambda(\theta),$$

*where $\Lambda(\theta)$ is defined in Equation (4).*

We now present binary stable regret lower bounds derived for some special instances.

**Corollary 5.3** (Serial Dictatorship). *Consider a general serial dictatorship instance $\theta$ with two sided uncertainty, where $N \leq K$, $\gamma_{j,i}(\theta) > \gamma_{j,i'}(\theta)$ for all $i < i'$ and $j$, and the unique stable matching is $\{(i,i) : i \in [N]\}$. Then the binary stable regret for $\theta$ is lower bounded as for any uniformly good policy $\pi$ is lower bounded as*

$$\liminf_{T \to \infty} \frac{\mathbb{E}[R^\pi_{0/1}(T; \theta)]}{\log(T)} \geq \max\left(\max_{i \in [N]} c_{u,i}(\theta), \max_{j \in [K]} c_{a,j}(\theta)\right)$$

*where,*

$$c_{u,i}(\theta) = \sum_{j \in D_i(\theta)} \frac{1}{kl(\gamma_{j,i}(\theta), \gamma_{j,j}(\theta))} + \sum_{j > i} \frac{1}{kl(\mu_{i,j}(\theta), \mu_{i,i}(\theta))},$$

---

derlying (unique) full rank is compatible to the partial rank under consideration. More generally, we may replace $(F_u, F_a)$ with $\theta$.

| Algorithm | Two sided | Market Assumptions | Upper Bound per User/Arm |
|---|---|---|---|
| C-UCB (Liu et al., 2020) | NO | Centralized | Pessimal $O(NK \log(T)/\Delta_{\min}^2)$ |
| UCB-D3 (Sankararaman et al., 2021) | NO | Serial Dictatorship | Unique $O(NK \log(T)/\Delta_{\min}^2)$ |
| CA-UCB (Liu et al., 2021) | NO | (user, arm) broadcast | Pessimal $O(\exp(N)N^5K^2 \log(T)/\Delta_{\min}^2)$ |
| UCB-D4 (Basu et al., 2021) | NO | uniqueness consistency | Unique $O(NK \log(T)/\Delta_{\min}^2)$ |
| UCB-DMA (Maheshwari et al., 2022) | NO | $\alpha$-reducible | Unique $O(\mathcal{C}_\alpha NK \log(T)/\Delta_{\min}^2)$ |
| ETGS (Kong & Li, 2023) | NO | (user, arm) broadcast | Optimal $O(K \log(T)/\Delta_{\min}^2)$ |
| PCA-DAA (Pokharel & Das, 2023) | YES | (user, arm) broadcast | - |
| ETGS+BB (Pagare & Ghosh, 2024) | YES | (user, arm) broadcast | User Optimal $O(K \log(T)/\Delta_{\min}^2))$ |
| CA-ETC (Pagare & Ghosh, 2024) | YES | no broadcast | User Optimal $O(poly(T))$ |
| Ours | YES | 2-bit broadcast | Pessimal & Binary $O(K \log(T)/\Delta_{\mathcal{A}}^2)$ |

| Two sided | Market Type | Lower Bound per User/Arm | |
|---|---|---|---|
| NO | No Broadcast | Unique $\Omega(N \log(T)/\Delta_{\min}^2 + K \log(T)/\Delta_{\min})$ (Sankararaman et al., 2021) | |
| YES | Centralized | Binary $\Omega(L \log(T)/\Delta_{\mathcal{A},\mathrm{avg}}^2)$ [Ours] | |

Table 1: A more comprehensive regret comparison. We consider three type of gaps satisfying $\Delta_{\min} \leq \Delta_{\mathcal{A}} \leq \Delta_{\mathcal{A},\mathrm{avg}}$. $C_\alpha$ is a parameter specific to $\alpha$-irreducibility in (Maheshwari et al., 2022). For lower bound, $L$ denotes the average number of (user, arm) pairs not participating in a stable matching with $(K - N) \leq L \leq (K - 1)$ (see Theorem 5.2). In regret upper bound, 'Unique' means unique stable matching, 'Optimal' means User optimal stable regret, 'Pessimal' means User pessimal stable regret, and 'Binary' means binary stable regret.

$$c_{a,j}(\theta) = \sum_{i:j \in D_i(\theta)} \frac{1}{kl(\gamma_{j,i}(\theta), \gamma_{j,j}(\theta))} + \sum_{i<j} \frac{1}{kl(\mu_{i,j}(\theta), \mu_{i,i}(\theta))}.$$

Note the above instance holds with out loss of generality up to renaming of arms and user, and thus covers all instances of general serial dictatorship. This lower bound differs from Sankararaman et al. (2021) in several ways. It's for binary stable regret (not per-user regret), applies to general serial matching (not just OSB instances), and accounts for two-sided (not just user-sided) uncertainty. Thus, the bounds are incomparable.

**Corollary 5.4** (General Instance with Redundant Arm). *Consider a general two-sided stable matching instance* $\theta$ *with redundant arms* $N < K$, *where* $j^*(i) \triangleq \max_j\{\mu_{i,j}(\theta) : (i,j) \in Locked(\theta)\}$. *Then the binary stable regret for* $\theta$ *is lower bounded as for any* uniformly good *policy* $\pi$ *is lower bounded as* $\liminf_{T \to \infty} \frac{\mathbb{E}[R_{0/1}^\pi(T;\theta)]}{\log(T)} \geq$

$$\max\left(\max_{i \in [N]} c_{u,i}(\theta), \max_{j:(\cdot,j) \notin Locked(\theta)} c_{a,j}(\theta)\right) \text{ where,}$$

$$c_{u,i}(\theta) = \sum_{j:(\cdot,j) \notin Locked(\theta)} \frac{1}{kl(\mu_{i,j}(\theta), \mu_{i,j^*(i)}(\theta))},$$

$$c_{a,j}(\theta) = \sum_{i \in [N]} \frac{1}{kl(\mu_{i,j}(\theta), \mu_{i,j^*(i)}(\theta))}.$$

**General Instances:** Because of space constraints, detailed definitions and results for the fully general case (where N may equal K) are relegated to Appendix G. We provide a brief and high-level overview here.

A *cover* of the set $\mathcal{A}(\theta)$ is a set of triplets (2nd and 3rd entries are unordered) $(i, \{j, j'\})$, $(j, \{i, i'\})$ for $i, i' \in$ $[N], j, j' \in [K]$ such that - (i) for each $(P_u, P_a) \in \mathcal{A}(\theta)$ there is at least one $(i, \{j, j'\})$ such that $j \underset{P_{u,i}}{\neq} j'$ or $(j, \{i, i'\})$ such that $i \underset{P_{a,j}}{\neq} i'$; (ii) at most one of $(i, j)$ or $(i, j')$ lies in $Locked(\theta)$, same for $(i, j)$ and $(j, i')$; and (iii) all the triplets are 'realizable'. See, Def. F.5 in appendix for formal version. Also, two triplets above are overlapping if they share a common (user, arm) pair, e.g. $(i, \{j, j'\})$ and $(j, \{i, i'\})$. We define a collection of 'non-connected' cover-group where no pair of covers share any overlapping triplet. See Def. F.6 in appendix. The set of cover-group for $\theta$ is $\mathcal{CG}(\theta)$.

Through a dual formulation of optimization problem in Theorem 5.2 we show in Theorem F.7 that the value $c(\theta)$ therein admits the lower bound

$$c(\theta) \geq \max_{G \in \mathcal{CG}(\theta)} \sum_{C \in G} kl(\theta, C)^{-1}, \quad (5)$$

where the divergence for a cover $C$ is given as

$$kl(\theta, C) \triangleq \sum_{i \in [N]} \max_{\substack{(i,j') \in Locked(\theta), \\ (i,\{j,j'\}) \in C}} kl(\mu_{ij}(\theta), \mu_{ij'}(\theta))$$

$$+ \sum_{j \in [K]} \max_{(i',j) \in Locked(\theta), (j,\{i,i'\}) \in C} kl(\gamma_{ji}(\theta), \gamma_{ji'}(\theta)).$$

The quantity $\mathcal{CG}(\theta)$ and $kl(\theta, C)$ depends on the combinatorial structure of the admissible partial rank set $\mathcal{A}(\theta)$, and any further simplification without large sub-optimality proves to be difficult.

**Remarks on Lower Bound:** Multiple remarks on our regret lower bound is in order.

◇ **OSSB Applicability:** We note that in Combes et al. (2015) the authors provide an algorithm that can attain the

regret lower bound named OSSB. However, as the size of the set of all matchings is exponential in $N$) a OSSB style algorithm is computationally prohibitive for our problem. More importantly, OSSB algorithm is designed for instances with unique optimal solution, so it is unclear if OSSB can be applied in our case directly.

$\diamond$ **Regret 'Free' Instances:** If for each pair $(i, j)$ there exists a stable match $\mathcal{M} \in \text{Stable}(\theta)$ then regret is $o(\log(T))$. Playing the stable matching in a round robin manner resolves all the gaps with $\exp(-\Omega(T))$ probability without any regret. So statistically $O(1)$ regret is feasible even for uniformly good policies. We present a detailed argument, and a sequence of such zero regret instances for all $N \geq 2$ in the appendix G. For $N = 2$ in the instance $U_1 : [1, 2], U_2 : [2, 1], A_1 : [2, 1], A_2 : [1, 2]$ both $[(1, 1), (2, 2)]$ and $[(2, 1), (1, 2)]$ are stable matching, and these two covers all edges.

# 6. Conclusion and Future Work

This work investigates two-sided bandit learning in general matching markets, providing both centralized and decentralized algorithms with logarithmic regret guarantees (Theorems 3.5 and 4.1, respectively). Our algorithms leverage the concept of super-stable matching (Irving, 1994), playing such a match when possible and exploring otherwise. Our centralized regret lower bound (Theorem F.7) highlights the importance of the super-stable set in determining the difficulty of stable matching bandit learning. While our current work employs a sub-optimal round-robin exploration strategy, future work will focus on refining exploration techniques to achieve tight regret bounds. Furthermore, while we guarantee pessimal stable regret, developing algorithms that optimize for specific objectives (e.g., user-optimal or social-optimal) represents another important direction. We hypothesize that our initial stable matching algorithm could be combined with the distributive lattice structure of the stable matching set to efficiently explore and identify optimal stable matching.

# Impact Statement

This paper presents a theoretical exploration in the field of Bandit learning in Matching markets with two sided uncertainty. The ideas presented may have future implications in developing efficient algorithms in various fields, such as e-commerce and crowd-sourcing marketplaces, ride-sharing systems. However, given the pure theoretical nature of this work, and gaps from practice (e.g. lack of contextual rewards) it is hard to foresee any specific impact at this stage.

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

## A. Related Work

**Stable Matching with Partial Preference:** Stable matching has been a successful concept for studying equilibria among incentivized agents in two-sided matching markets. In the foundational work Gale & Shapley (1962), participants express preferences through strict rankings. A natural extension allows for more realistic scenarios with partial preferences. Irving (1994) expanded the concept of stability to partial preferences by introducing three distinct stability concepts: weak, strong, and super-stability. Our work draws from Irving (1994), and connects super-stability to the inherent hardness of learning stable mathcing under bandit feedback. The Extended Gale-Shapley algorithm there is the central algorithm that we use breaking from the literature that exclusively uses the original Gale-Shapley algorithm. Next, we build on the work of Spieker (1995) which provides structure of the set of super-stable matchings.

**Bandit Learning in Matching Markets:** The work of (Das & Kamenica, 2005) introduced the problem of learning stable matching for a matching markets with one-sided uncertainty, i.e. the users have unknown preferences but the arms have complete knowledge of their prefrences. In (Liu et al., 2020) the authors provided a UCB based centralized algorithm with the first theoretical regret guarantee of $O(NK \log(T)/\Delta_{\min}^2)$. An explore-then-commit algorithm with the knowledge of minimum gap $\Delta_{\min}$ was also proposed. Next, the work was extended to the decentralized setup that removes the knowledge of gap but imposes uniqueness of stable matching via structural assumptions on preference rank (Sankararaman et al., 2021; Basu et al., 2021; Maheshwari et al., 2022). The algorithms proposed therein achieve a $O(NK \log(T)/\Delta_{\min}^2)$ user pessimal stable regret. The collision avoidance UCB based algorithm proposed in (Liu et al., 2021) achieves $O(\exp(N) \log^2(T)/\Delta_{\min}^2)$ regret in general matching markets.

A separate thread of work explored the Explore-then-commit type algorithms for general markets starting from (Basu et al., 2021), and improved in the ETGS (Kong & Li, 2023). The ETGS type algorithms are able to achieve a $O(K \log(T)/\Delta_{\min}^2)$ user optimal stable regret. A more recent work (Hosseini et al., 2024) provides sample complexity bounds, which translates to $O(E \log(T)/\Delta_{\min}^2)$ regret bound, using a concept of envy-set where $E$ is the size of average envy set, and may be less than $K$ based on the underlying instances. There has been multiple works with ETGS as the core where a user can match with multiple arms (upto a quota) in each round (Kong & Li, 2024; Saha et al., 2024).

Only a select few recent works (Pokharel & Das, 2023; Pagare & Ghosh, 2024; Zhang & Fang) study matching markets with two-sided uncertainty where both sides need to learn their respective preferences from bandit feedback. In (Pagare & Ghosh, 2024), the authors present a black-board model similar to ours and attain a $O(K \log(T)/\Delta_{\min}^2)$ algorithm similar to centralized ETGS mentioned in this work. Similar guarantees are provided in (Zhang & Fang) without baclkboard, while (Pokharel & Das, 2023) lacks any theoretical guarantee. But all these works focus on user optimal stable match learning. This is incomparable to the task in this paper of finding any one stable matching.

Our work provides an entirely new algorithmic idea by applying the Extended Gale Shapley algorithm (Irving, 1994). Furthermore, we provide a new instance dependent lower bound for general instances for centralized setting, and hence also for decentralized setting. The long standing regret lower bound before this work comes from (Basu et al., 2021) which is applicable only to decentralized setting, and for the limited special case of serial dictatorship. Our work is based on

On the algorithmic side, apart from Gale Shapley based bandit algorithms, the authors in (Etesami & Srikant, 2024) take a new game-theoretic approach of preference learning with exponential weight learning with one-sided uncertainty and provide a $O(N^2 log^3(T) + N^2 log(T)/\Delta_{\min}^N)$ regret bounds. Also, there has been work that involved non-stationarity in the mean rewards of the arms (Ghosh et al., 2024; Maheshwari et al., 2022), and mean rewards have a linear contextual structure (Jagadeesan et al., 2021; Parikh et al., 2024). Finally, in (Lin et al., 2024) the authors study approximation ratios and learning to find weak-stable matches which is different from the standard setup.

## B. Experiments

In this section, we numerically study the behavior of the proposed centralized algorithm. As the decentralized algorithm is guaranteed to have only a regret $O(N^2)$ away from the centralized one, we omit the decentralized algorithm. We first show the pessimal stable regret dynamics of the centralized algorithm for a system with $N$ users and $K$ arms. We generate a bandit instance by selecting a random preference order for each user and each arm. We generate the mean rewards $\mu_{i,j}$ and $\gamma_{j,i}$, for $i \in [N]$ and $j \in [K]$ such that the permutation is randomized, and the gaps are chosen uniformly at random from $[\Delta_{\min}, \Delta_{\max}]$. For each reward observation we apply independent Gaussian noise with standard deviation $\sigma$ (the UCB and LCB bonuses are also scaled with $\sigma$). We plot the evolution of the mean cumulative regret with time for our proposed

Algorithm 1, and a centralized Explore-then-Gale Shapley (ETGS) algorithm of Kong et al. (Kong & Li, 2023) for fair comparison.[6] Centralized ETGS explores until the top $N$ ranks are resolved for each user and arms, and then commits to the Gale Shapley solution afterwards.

In Figure 1, we consider one such instance with $N = 8$ users, $M = 8$ arms, $\Delta_{\min} = 0.2$, $\Delta_{\max} = 0.5$, and $\sigma = 0.4$. We observe the logarithmic pessimal stable regret for both the algorithms. However, clearly the proposed Extended-GS based algorithm outperforms the Centralized ETGS adapted from Kong et al. (Kong & Li, 2023). We plot the mean rewards, as well as $25\%$ and $75\%$ regret plots averaged over 20 sample paths. Next, in Figrue 2 we study the dependence on $\sigma/\Delta_{avg}$ by varying $\sigma$ for $N = K = 5$ and $\Delta_{\min} = 0.2$, $\Delta_{\max} = 0.5$. We observe that there is a clear separation between the Centralized ETGS and proposed Extended-GS algorithm showing the superiority of the latter. Both the algorithms initially explore but Extended-GS algorithm finds an admissible partial rank much earlier than the top $N$ rank resolution leading to a lower regret than ETGS. Figrue 3 exhibits similar behavior with varying gaps $\Delta_{\min}$ and $\Delta_{\max}$.

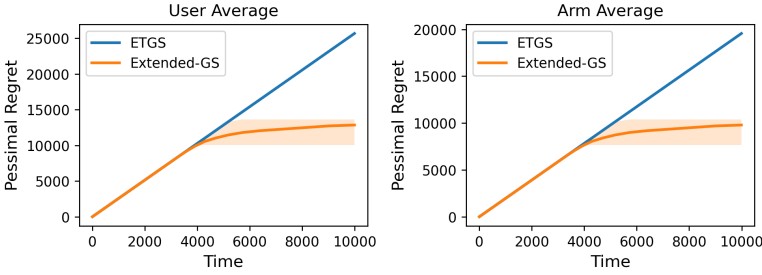

Figure 1: Pessimal Stable Regret for $N = 8$ Users, $M = 8$ Arms, $\Delta_{\min} = 0.2$, $\Delta_{\max} = 0.5$, and $\sigma = 0.5$.

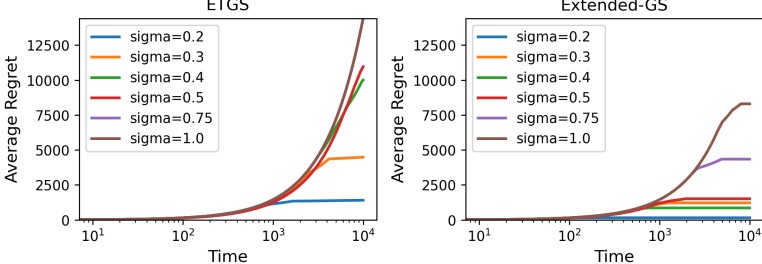

Figure 2: Average pessimal stable regret across all users and arms with different $\sigma$, with $N = 5$ Users, $M = 5$ Arms, $\Delta_{\min} = 0.2$, $\Delta_{\max} = 0.5$.

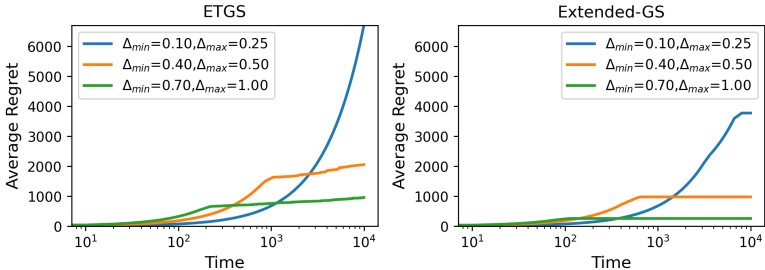

Figure 3: Average pessimal stable regret across all users and arms with different $\Delta_{\min}$, and $\Delta_{\max}$, with $N = 5$ Users, $M = 5$ Arms, $\sigma = 0.4$.

---

[6] A decentralized version of this algorithm with shared global flags (aka blackboard) is presented in Algorithm 2 in (Pagare & Ghosh, 2024).

## C. Algorithms Addendum

In round $t$ for $i \in [N]$ and $j \in [K]$, the estimated mean rewards $\hat{\mu}_{i,j}(t)$ and $\hat{\gamma}_{j,i}(t)$, and number of samples $n_{i,j}(t)$ for our Algorithms (Algo 1,Algo 4, and Algo 3) are given as

$$n_{i,j}(t) = n_{i,j}(t-1) + \mathbb{1}((i,j) \in M(t))$$

$$\hat{\mu}_{i,j}(t) = \frac{n_{i,j}(t-1)\hat{\mu}_{i,j}(t-1) + Y_i(t)}{n_{i,j}(t-1) + 1} \text{ if } (i,j) \in M(t) \text{ else } \hat{\mu}_{i,j}(t-1)$$

$$\hat{\gamma}_{j,i}(t) = \frac{n_{i,j}(t-1)\hat{\gamma}_{j,i}(t-1) + \tilde{Y}_j(t)}{n_{i,j}(t-1) + 1} \text{ if } (i,j) \in M(t) \text{ else } \hat{\gamma}_{j,i}(t-1).$$

We now state the Extended GS algorithm from Irving (Irving, 1994).

---

**Algorithm 2:** Extended Gale Shapley (EXTENDED-GS)

---

1 **Input:** User partial rankings $\mathcal{P}_u$, and arm partial rankings $\mathcal{P}_a$
2 **Initialize:** $m_i = \emptyset$ for all $i \in [N]$
3 **while** *for all $i \in [N]$, $P_{u,i} \neq \emptyset \wedge \exists i \in [N], m_i = \emptyset$* **do**
4      **for** *each user $i \in [N]$* **do**
5          Propose to all 'source' arms $j \in [K]$ under partial rank $P_{u,j}$
6      **for** *arm $j \in [K]$* **do**
7          Gather the proposing users $Prop(j)$
8          Get the 'source' nodes $S(j)$ in the DAG $P_{a,j} \cap Prop(j)$
9          **for** *each user $k \underset{P_{a,j}}{<} s$, for some $s \in S(j)$* **do**
10              Remove arm $j$ from $P_{u,k}$, and user $k$ from $P_{a,j}$
11          **if** $|S(j)| = 1$ **then**
12              ▷ A user $i$ can be accepted by multiple arms
13              For $i_j^* \in S_j$, append $j$ to $m_{i_j^*}$
14          **else**
15              **for** *each user $k$ in the 'sink' of $P_{a,j}$ (a DAG)* **do**
16                  Remove arm $j$ from $P_{u,k}$, and user $k$ from $P_{a,j}$
17 **if** *for all $i \in [N]$, $m_i \neq \emptyset$* **then**
18      Return $M_{stable} = \{(i, m_i) : i \in [N]\}$
19 **else**
20      Return $\emptyset$

---

Next, we state the full decentralized algorithms here. We colorize the communication among the users and the arms, the observations, and the update to global shared flags.

---

**Algorithm 3:** Decentralized Two-Sided Matching Bandit, User $i$

---

1   **Global Flags:** $restart \leftarrow False$, $success \leftarrow False$
2   **Local Indices:** Exploration $\tau_{ex} \leftarrow 0$, Phase count $\tau_\ell \leftarrow 1$,
3            Phase exploration indicator $explore(1) \leftarrow False$
4   **Initial Ranking:** $P_{u,i}(0) \leftarrow$ empty $K$ node DAG
5   ▷ Let global time $t \leftarrow 0$ (convention)
6   **for** $\tau_\ell \geq 1$ **do**
7      Update $PR_{u,i}(\tau_\ell) \leftarrow P_{u,i}(t)$
8      Read shared flag $success$ and set $explore(\tau_\ell) \leftarrow \neg success$
9      **if** $success$ **then**
10         $M_i(\tau_\ell) \leftarrow \arg\max_{j:S_{i,j}(t)=accept} \mu\text{-}ucb_{i,j}(t)$
11      **while** $True$ **do**
12         ▷ Globally: $restart \leftarrow False$, $success \leftarrow True$
13         ▷ Increment global time $t \leftarrow t + 1$
14         Propose for all arms $j$ in 'source' of $PR_{u,i}(\tau_\ell)$, $\tilde{S}_j(t) \leftarrow \tilde{S}_j(t) \cup \{i\}$
15         Receive the signals from the proposed arms $S_{i,j}(t)$
16         Delete all $\{j : S_{i,j}(t) = \text{reject}\}$ from $PR_{u,i}(\tau_\ell)$
17         **if** $explore(\tau_\ell)$ **then**
18            Match with $m_i(t) \leftarrow (i + \tau_{ex}) \mod K$, and set $\tau_{ex} \leftarrow \tau_{ex} + 1$.
19         **else**
20            Match with $m_i(t) \leftarrow M_i(\tau_\ell)$.     ▷ Match from previous phase
21         ▷ No collision occurs, and $m_i(t) \neq \emptyset$
22         Observe reward $Y_{i,m_i(t)}(t)$ and update UCB and LCB using Eq. (1).
23         Compute $P_{u,i}(t)$ using Eq. (3).
24         ▷ GLOBAL COMMUNICATION
25         **if** $\forall$ *proposed arm* $j$, $S_{i,j}(t) = \text{reject}$ **then**
26            Set $success \leftarrow success \land False$
27         **else**
28            Set $success \leftarrow success \land True$
29         Read shared flag $success$ (once all updates are done)
30         **if** $PR_{u,i}(\tau_\ell) = \emptyset \lor success$ **then**
31            Set $restart \leftarrow restart \lor True$
32         **else**
33            Set $restart \leftarrow restart \lor False$
34         Read shared flag $restart$ (once all updates are done)
35         **if** $restart$ **then**
36            break ▷ Enters a new phase

---

The algorithm for the arm is given as follows.

---

**Algorithm 4:** Decentralized Two-Sided Matching Bandit, Arm $j$

---

1   **Global Flags:** $restart$
2   **Counter:** Phase counter $\tau_\ell \leftarrow 1$
3   **Initial Ranking:** $P_{a,j}(0) \leftarrow$ empty $N$ node DAG
4   ▷ Let global time $t \leftarrow 0$ (convention)
5   **for** $\tau_\ell \geq 1$ **do**
6      Update $PR_{a,j}(\tau_\ell) \leftarrow P_{a,j}(t)$
7      **while** $True$ **do**
8         ▷ Increment global time $t \leftarrow t + 1$
9         Receive proposals from users $\tilde{S}_j(t)$
10        Get 'source' nodes $I^*(j)$ from $\tilde{S}_j(t) \cap PR_{a,j}(\tau_\ell)$
11        **if** $|I^*(j)| = 1$ **then**
12           Accept the unique user $i^*(j) \in I^*(j)$, by setting $S_{i^*(j),j}(t) = $ accept
13           For all $i \in \tilde{S}_j(t)$ and $i \neq i^*(j)$, set $S_{i,j}(t) = $ reject
14           For all $i < i^*(j)$, delete $i$ in $PR_{a,j}(\tau_\ell)$
15        **else if** $|I^*(j)| > 1$ **then**
16           For all $i \in \tilde{S}_j(t)$, set $S_{i,j}(t) = $ reject
17           All 'tail' user $i$ in $PR_{a,j}(\tau_\ell)$ are deleted
18        ▷ Arm $j$ matches with user $m_j^{-1}(t)$
19        ▷ $m_j^{-1}(t) = \{i : (i + \tau_{ex}) \mod K = j\}$ (possibly $\emptyset$) if $explore(\tau_\ell) = True$
20        ▷ $m_j^{-1}(t) = \emptyset$ or $i^*(j)$ if $explore(\tau_\ell) = False$
21        **if** $m_j^{-1}(t) \neq \emptyset$ **then**
22           Observe reward $\tilde{Y}_{j,m_j^{-1}(t)}(t)$ and update UCB and LCB using Eq. (2)
23        Compute $P_{a,j}(t)$ using Eq. (3).
24        ▷ GLOBAL COMMUNICATION
25        Read shared flag $restart$ (once all users finish update)
26        **if** $restart$ **then**
27           break     ▷ Enters a new phase

---

## D. Proofs of Preliminary Section

*Proof of Lemma 3.4.* Given $(F_u, F_a) \in$ FullRank$(P_u, P_a)$, we need to show that SuperStable$(P_u, P_a) \cap$ Stable$(F_u, F_a) \neq \emptyset$ in order to prove this lemma. It is sufficient to argue that FullRank$(P_u, P_a) \subseteq$ FullRank$(P'_u, P'_a)$ for *any* $(P'_u, P'_a) \in \mathcal{A}(F_u, F_a)$, because

$$\text{Stable}(F_u, F_a) \cap \text{SuperStable}(P_u, P_a) \overset{(i)}{=} \text{Stable}(F_u, F_a) \cap \bigcap_{(F'_u, F'_a) \in \text{FullRank}(P_u, P_a)} \text{Stable}(F'_u, F'_a)$$

$$\overset{(ii)}{\supseteq} \text{Stable}(F_u, F_a) \cap \bigcap_{(F'_u, F'_a) \in \text{FullRank}(P'_u, P'_a)} \text{Stable}(F'_u, F'_a)$$

$$\overset{(iii)}{=} \text{Stable}(F_u, F_a) \cap \text{SuperStable}(P'_u, P'_a) \overset{(iv)}{\neq} \emptyset$$

The equality $(i)$ and $(iii)$ is by the definition of super-stable set. Inequality $(ii)$ follows from the assumption FullRank$(P_u, P_a) \subseteq$ FullRank$(P'_u, P'_a)$, and inequality $(iv)$ holds because we chose $(P'_u, P'_a) \in \mathcal{A}(F_u, F_a)$.

What is left to complete the proof is to show that the assumption FullRank$(P_u, P_a) \subseteq$ FullRank$(P'_u, P'_a)$ is true for some $(P'_u, P'_a) \in \mathcal{A}(F_u, F_a)$. Let us choose a partial rank $(P'_u, P'_a) \in \mathcal{A}(F_u, F_a)$ with a minimum gap $\Delta_\mathcal{A}(\boldsymbol{\mu}, \boldsymbol{\gamma})$ which exists due to the finiteness of the set $\mathcal{A}(F_u, F_a)$. For any pair of users $i, i' \in [N]$ and arms $j, j' \in [K]$, we have by definition of

overlap width, and because $\mathcal{W}_{\mathrm{ov}}(P_u, P_a; \boldsymbol{\mu}, \boldsymbol{\gamma}) < \Delta_{\mathcal{A}}(\boldsymbol{\mu}, \boldsymbol{\gamma})$

$$|\mu_{i,j} - \mu_{i,j'}| \geq \Delta_{\mathcal{A}}(\boldsymbol{\mu}, \boldsymbol{\gamma}) \implies j \underset{P_{u,i}}{\neq} j' \text{ and } |\gamma_{j,i} - \gamma_{j,i'}| \geq \Delta_{\mathcal{A}}(\boldsymbol{\mu}, \boldsymbol{\gamma}) \implies i \underset{P_{a,j}}{\neq} i'.$$

However, by definition of $\Delta_{\mathcal{A}}(\boldsymbol{\mu}, \boldsymbol{\gamma})$ and the fact that $(P_u', P_a')$ attains the maximum gap we know

$$j \underset{P_{u,i}'}{\neq} j' \implies |\mu_{i,j} - \mu_{i,j'}| \geq \Delta_{\mathcal{A}}(\boldsymbol{\mu}, \boldsymbol{\gamma}) \implies j \underset{P_{u,i}}{\neq} j'$$

$$i \underset{P_{a,j}}{\neq} i' \implies |\gamma_{j,i} - \gamma_{j,i'}| \geq \Delta_{\mathcal{A}}(\boldsymbol{\mu}, \boldsymbol{\gamma}) \implies i \underset{P_{a,j}}{\neq} i'.$$

In other words, any pair of users (or arms) that are unequal in the partial rank $(P_u', P_a')$ are also unequal in the partial rank $(P_u, P_a)$. This in turn implies that $\mathrm{FullRank}(P_u, P_a) \subseteq \mathrm{FullRank}(P_u', P_a')$. This completes our proof. $\qquad\square$

## E. Proofs of Regret Upper Bounds

We define the good event $\mathcal{G}_t$ for all time $t \geq 1$ as

$$\forall i \in [N], j \in [K], \max\left(|\mu_{i,j} - \hat{\mu}_{i,j}(t)|, |\gamma_{j,i} - \hat{\gamma}_{j,i}(t)|\right) \leq \sqrt{\frac{6\log(t)}{n_{i,j}(t)}}.$$

**Lemma E.1.** *The probability of 'bad' event $\mathcal{G}_t^c$, for any $t \geq 1$, satisfies $\mathbb{P}(\mathcal{G}_t^c) \leq 2NK/t^2$.*

*Proof.* We start our bound as follows,

$$\mathbb{P}(\mathcal{G}_t^c)$$

$$= \mathbb{P}\left(\max_{i \in [N], j \in [K]} \max\left(|\mu_{i,j} - \hat{\mu}_{i,j}(t)|, |\gamma_{j,i} - \hat{\gamma}_{j,i}(t)|\right) > \sqrt{\frac{6\log(t)}{n_{i,j}(t)}}\right)$$

$$\leq \sum_{i \in [N], j \in [K]} \mathbb{P}\left(|\mu_{i,j} - \hat{\mu}_{i,j}(t)| > \sqrt{\frac{6\log(T)}{n_{i,j}(t)}}\right) + \sum_{i \in [N], j \in [K]} \mathbb{P}\left(|\gamma_{j,i} - \hat{\gamma}_{j,i}(t)| > \sqrt{\frac{6\log(t)}{n_{i,j}(t)}}\right)$$

$$\leq \sum_{i \in [N], j \in [K]} \sum_{s=1}^{t} \left(\mathbb{P}\left(n_{i,j}(t) = s, |\mu_{i,j} - \hat{\mu}_{i,j}(t)| > \sqrt{\frac{6\log(t)}{s}}\right) + \mathbb{P}\left(|\gamma_{j,i} - \hat{\gamma}_{j,i}(t)| > \sqrt{\frac{6\log(t)}{s}}\right)\right)$$

$$\overset{(i)}{\leq} 2NK \sum_{s=1}^{t} \mathbb{P}\left(|\hat{\mu}_X^{(s)}| > \sqrt{\frac{6\log(t)}{s}} \ \Big| \ n_{i,j}(t) = s\right)$$

$$\leq 2NK \sum_{s=1}^{t} \exp(-3\log(t)) \leq 2NK/t^2.$$

All the rewards come from some 1-sub-gaussian distribution. Therefore, we can bound inequality $(i)$ using the sub-gaussian concentration bound with $m$ samples for a 1-subgaussian zero mean distribution $X$ (c.f. Lattimore et al ()) $\mathbb{P}(|\hat{\mu}_X^{(s)}| > \epsilon) \leq 2\exp(-s\epsilon^2/2)$. $\qquad\square$

**Lemma E.2.** *Conditioned on the good event $\mathcal{G}(t)$, the recovered set of partial matching $(\mathcal{P}_u(t), \mathcal{P}_a(t))$ satisfies that $(F_u^*, F_a^*) \in FullRank((\mathcal{P}_u(t), \mathcal{P}_a(t)))$.*

*Proof.* We know that given the good event $\mathcal{G}(t)$, for any two users $i, i' \in [N]$ and any two arm $j \in [K]$ we have

$$\mu\text{-}ucb_{i,j}(t) \geq \mu_{i,j} + \sqrt{\tfrac{6\log(T)}{n_{i,j}(t)}} - |\hat{\mu}_{i,j} - \mu_{i,j}| \geq \mu_{i,j},$$

$$\mu\text{-}lcb_{i,j}(t) \leq \mu_{i,j} - \sqrt{\tfrac{6\log(T)}{n_{i,j}(t)}} + |\hat{\mu}_{i,j} - \mu_{i,j}| \leq \mu_{i,j}.$$

Therefore, $\mu\text{-}ucb_{i,j'}(t) < \mu\text{-}lcb_{i,j}(t)$ implies that $\mu_{i,j} > \mu_{i,j'}$ under the good event $\mathcal{G}(t)$. Similarly, $\gamma\text{-}ucb_{i',j}(t) < \gamma\text{-}lcb_{i,j}(t)$ implies $\gamma_{i,j} > \gamma_{i',j}$ under the good event $\mathcal{G}(t)$.

Therefore, for any fixed user $i \in [N]$ any pair of arms $j, j' \in [K]$ if we have $\mu_{i,j} > \mu_{i,j'}$, i.e. $j \underset{F_{u,i}^*}{>} j'$, then $j \underset{\mathcal{P}_{u,i}(t)}{\geq} j'$.

Similarly, for any fixed arm $j \in [K]$ any pair of arms $i, i' \in [K]$, $i \underset{F_{a,j}^*}{>} i'$ implies $i \underset{\mathcal{P}_{u,j}(t)}{\geq} i'$. This implies by definition that $(F_u^*, F_a^*) \in FullRank((\mathcal{P}_u(t), \mathcal{P}_a(t)))$. $\qquad\square$

**Lemma E.3** (Convergence to Admissible Set). *Conditioned on the good event $\cap_{t \geq 1} \mathcal{G}(t)$, for any time $t \geq 1$ if number of explorations $N_{explore}(t) > K + \frac{96K \log(T)}{\Delta_{\mathcal{A}}^2(\boldsymbol{\mu}, \boldsymbol{\gamma})}$ we have $(\mathcal{P}_u(t), \mathcal{P}_a(t)) \in \mathcal{A}(F_u^*, F_a^*)$, i.e. the recovered partial rank lies in the set of admissible partial rank instances of the true rankings.*

*Proof of Lemma E.3.* Let $N_{explore}(t)$ be the number of exploration steps taken by the centralized platform up to time $t$. That implies that for each user and arm pair $(i, j)$ for $i \in [N]$ and $j \in [K]$, the number of samples $n_{i,j}(t) \geq (N_{explore}(t) - K)/K$. With $n_{i,j}(t)$ many samples, under the good event $\mathcal{G}(t)$ by definition

$$\max\left(|\mu_{i,j} - \hat{\mu}_{i,j}(t)|, |\gamma_{j,i} - \hat{\gamma}_{j,i}(t)|\right) \leq \sqrt{\frac{6 \log(t)}{n_{i,j}(t)}}.$$

Therefore, for any fixed user $i \in [N]$ and any pair of $j, j' \in [K]$, if

$$(\mu_{i,j} - \mu_{i,j'}) > 4\sqrt{\frac{6K \log(T)}{N_{explore}(t) - K}} \geq 2\sqrt{6 \log(T)}\left(\frac{1}{\sqrt{n_{i,j}(t)}} + \frac{1}{\sqrt{n_{i,j'}(t)}}\right),$$

then we have $\mu\text{-}lcb_{i,j}(t) > \mu\text{-}ucb_{i,j'}(t)$ and $j \underset{\mathcal{P}_{u,i}(t)}{>} j'$. Similarly, if $(\gamma_{j,i} - \gamma_{j,i'}) > 4\sqrt{\frac{6K \log(T)}{N_{explore}(t) - K}}$ we have $i \underset{\mathcal{P}_{a,j}(t)}{>} i'$.

Therefore, by definition $\mathcal{W}_{\text{ov}}(\mathcal{P}_u(t), \mathcal{P}_a(t); \boldsymbol{\mu}, \boldsymbol{\gamma}) \leq 4\sqrt{\frac{6K \log(T)}{N_{explore}(t) - K}}$.

From Lemma E.2 $((F_u^*, F_a^*) \in \text{FullRank}(\mathcal{P}_u(t), \mathcal{P}_a(t))$ under the good event $\mathcal{G}_t$. Therefore, under good event $\cap_t \mathcal{G}_t$, from Lemma 3.4 if $\mathcal{W}_{\text{ov}}(\mathcal{P}_u(t), \mathcal{P}_a(t); \boldsymbol{\mu}, \boldsymbol{\gamma}) < \Delta_{\mathcal{A}}(\boldsymbol{\mu}, \boldsymbol{\gamma})$ then $(\mathcal{P}_u(t), \mathcal{P}_a(t)) \in \mathcal{A}(F_u^*, F_a^*)$. Therefore, for all $t$ such that

$$N_{explore}(t) \geq K + 96K \frac{\log(T)}{\Delta_{\mathcal{A}}^2(\boldsymbol{\mu}, \boldsymbol{\gamma})}$$

we have $(\mathcal{P}_u(t), \mathcal{P}_a(t)) \in \mathcal{A}(F_u^*, F_a^*)$. $\qquad\square$

**Lemma E.4.** *Conditioned on the good event $\cap_{t \geq 1} \mathcal{G}(t)$, for some $t \geq 1$ if and only if $(\mathcal{P}_u(t), \mathcal{P}_a(t)) \notin \mathcal{A}(F_u^*, F_a^*)$ then exploration is triggered in round $t$.*

*Proof.* For round $t \geq 1$, if Algorithm 2 does not return a stable match then exploration is triggered in Algorithm 1. Therefore, we need to show that under $\cap_{t \geq 1} \mathcal{G}(t)$, if $(\mathcal{P}_u(t), \mathcal{P}_a(t)) \notin \mathcal{A}(F_u^*, F_a^*)$ then Algorithm 2 does not return a stable match. Note that from Lemma E.2 we have $(F_u^*, F_a^*) \in \text{FullRank}(\mathcal{P}_u(t), \mathcal{P}_a(t))$ under the good event $\mathcal{G}_t$. If there is a stable match returned by Algorithm 2 in round $t$, then $\text{SuperStable}(\mathcal{P}_u(t), \mathcal{P}_a(t)) \neq \emptyset$. Therefore, by definition $(\mathcal{P}_u(t), \mathcal{P}_a(t)) \in \mathcal{A}(F_u^*, F_a^*)$. This proves the lemma. $\qquad\square$

***Proof of Centralized Upper Bound Theorem 3.5.*** We now create the regret decomposition. We first define the following for notational simplicity

$$Th(T) = K + 96K \frac{\log(T)}{\Delta_{\mathcal{A}}^2(\boldsymbol{\mu}, \boldsymbol{\gamma})},$$

where $Th(T)$ corresponds to the sufficient amount of exploration necessary to find a super-stable matching for any $t \leq T$ (we will show this shortly).

$\mathbb{E}[R_{u,i}(T)](\mu_{i,pess} - \mu_{i,\min})^{-1}$

$$\overset{(i)}{\leq} \mathbb{E}\Big[\sum_{t=1}^{T} \mathbb{1}\big(M(t) \notin \text{Stable}(F_u^*, F_a^*)\big)\Big]$$

$$\overset{(ii)}{=} \mathbb{E}\Big[\sum_{t=1}^{T} \mathbb{1}\big((\mathcal{P}_u(t), \mathcal{P}_a(t)) \notin \mathcal{A}(F_u^*, F_a^*)\big)\Big]$$

$$\overset{(iii)}{\leq} \mathbb{E}\Big[\sum_{t=1}^{T} \mathbb{1}\big((\mathcal{P}_u(t), \mathcal{P}_a(t)) \notin \mathcal{A}(F_u^*, F_a^*)\big)\big|\cap_t \mathcal{G}_t\Big] + \sum_{t=1}^{T} \mathbb{P}(\mathcal{G}_t^c)$$

$$\overset{(iv)}{\leq} \mathbb{E}\Big[\sum_{t=1}^{T} \mathbb{1}\big((\mathcal{P}_u(t), \mathcal{P}_a(t)) \notin \mathcal{A}(F_u^*, F_a^*)\big)\mathbb{1}\big(N_{explore}(T) \leq Th(T)\big)\big|\cap_t \mathcal{G}_t\Big]$$

$$+ \mathbb{E}\Big[\sum_{t=1}^{T} \mathbb{1}\big((\mathcal{P}_u(t), \mathcal{P}_a(t)) \notin \mathcal{A}(F_u^*, F_a^*)\big)\mathbb{1}\big(N_{explore}(T) > Th(T)\big)\big|\cap_t \mathcal{G}_t\Big] + \sum_{t=1}^{T} 2NK/t^2$$

$$\overset{(v)}{\leq} Th(T) + T\mathbb{P}\Big[N_{explore}(T) > Th(T)\big|\cap_t \mathcal{G}_t\Big] + NK\pi^2/3$$

$$\overset{(vi)}{\leq} Th(T) + T\mathbb{P}\Big[\exists 1 \leq s \leq T : N_{explore}(s) = Th(T) \wedge \mathbb{1}\big((\mathcal{P}_u(s), \mathcal{P}_a(s)) \notin \mathcal{A}(F_u^*, F_a^*)\big)\big|\cap_t \mathcal{G}_t\Big] + NK\pi^2/3$$

$$\overset{(vii)}{\leq} Th(T) + NK\pi^2/3$$

The first inequality $(i)$ simply states that we incur regret $(\mu_{i,pess} - \mu_{i,\min})$ regret if Algorithm 1 does not create a stable matching $M(t)$ at round $t$. Inequality $(ii)$, follows as $(\mathcal{P}_u(t), \mathcal{P}_a(t)) \in \mathcal{A}(F_u^*, F_a^*)$ implies $M(t)$ is a stable matching under true preference by definition of $\mathcal{A}(F_u^*, F_a^*)$. In equality $(iii)$ we condition under good event $\cap_t\mathcal{G}_t$, and it's complement $\cup_t\mathcal{G}_t^c$. We also use a union bound over $t$, to upper bound the latter quantity by $(\mu_{i,pess} - \mu_{i,\min})\sum_{t=1}^{T}\mathbb{P}(\mathcal{G}_t^c)$. In equality $(iv)$ we further break the first term by considering two cases: (a) when $(N_{explore}(T) > Th(T))$ and (b) it's complement. Inequality $(v)$ first bounds the case where $N_{explore}(T) \leq Th(T)$. From Lemma E.4 it follows that under good event $\cap_t\mathcal{G}_t$, $N_{explore}(T) = \sum_{t=1}^{T}\mathbb{1}\big((\mathcal{P}_u(t), \mathcal{P}_a(t)) \notin \mathcal{A}(F_u^*, F_a^*)\big)$. Hence, under $\cap_t\mathcal{G}_t$, $N_{explore}(T) \leq Th(T)$ gives a regret bound of $(\mu_{i,pess} - \mu_{i,\min})Th(T)$. Inequality $(v)$ next bounds the second term with a trivial $T$ bound on $\sum_{t=1}^{T}\mathbb{1}\big((\mathcal{P}_u(t), \mathcal{P}_a(t)) \notin \mathcal{A}(F_u^*, F_a^*)\big)$. In $(vi)$, we notice that in order $N_{explore}(T) > Th(T)$ to hold there must be a time $s$ when $N_{explore}(T) = Th(T)$ and $(\mathcal{P}_u(s), \mathcal{P}_a(s)) \notin \mathcal{A}(F_u^*, F_a^*)$. Finally, due to Lemma E.3 we know that this event is not possible which gives our final bound.

A similar chain of inequalities proves the regret bound for $\mathbb{E}[R_{a,j}(T)]$. $\qquad\square$

***Proof of Decentralized Upper Bound Theorem 4.1.*** We now create the regret decomposition. We first define the following for notational simplicity

$$Th'(T) = 1 + N^2 + Th(t) = 1 + N^2 + K + 96K\frac{\log(T)}{\Delta_{\mathcal{A}}^2(\boldsymbol{\mu}, \boldsymbol{\gamma})},$$

where $Th'(T)$, similar to the centralized case, corresponds to the sufficient amount of exploration necessary to find a super-stable matching for any $t \leq T$ (we will show this shortly). Note that as our exploration may cross the centralized threshold at the beginning of a phase, it can occur for $N^2$ extra rounds.

We note that the *restart* flag ensures synchronization of each phase across the agents and arms. Each phase of the decentralized system follows one round of centralized system. Hence, the following results hold due to a similar line of reasoning as in the centralized system.

1. If and only if $(PR_u(\tau_\ell), PR_a(\tau_\ell)) \in \mathcal{A}(F_u^*, F_a^*)$ then the phase $\tau_\ell$ ends in 'success' and the recovered mapping used in phase $(\tau_\ell + 1)$, $M(\tau_\ell + 1) \in \text{Stable}(F_u^*, F_a^*)$. This is true by definition of $\mathcal{A}(F_u^*, F_a^*)$.

We now present the on the regret decomposition of the user $i$ below.

$$\mathbb{E}[R_{u,i}(T)](\mu_{i,pess} - \mu_{i,\min})^{-1}$$

$$\overset{(i)}{\leq} \mathbb{E}\Big[\sum_{t=1}^{T} \mathbb{1}\big(M(t) \notin \text{Stable}(F_u^*, F_a^*)\big)\Big]$$

$$\overset{(ii)}{=} \mathbb{E}\Big[\sum_{\tau_\ell \geq 1} \sum_{t \in Phase(\tau_\ell)} \mathbb{1}\big(explore(\tau_\ell)\big)\Big]$$

$$\overset{(iii)}{\leq} \mathbb{E}\Big[\sum_{\tau_\ell \geq 1} n(\tau_\ell)\mathbb{1}\big(explore(\tau_\ell)\big)\big|\cap_t \mathcal{G}_t\Big] + \sum_{t=1}^{T}\mathbb{P}(\mathcal{G}_t^c)$$

$$\overset{(iv)}{\leq} \mathbb{E}\Big[\sum_{\tau_\ell \geq 1} n(\tau_\ell)\mathbb{1}\big(explore(\tau_\ell)\big)\mathbb{1}\big(N_{explore}(T) \leq Th'(T)\big)\big|\cap_t \mathcal{G}_t\Big]$$

$$+ \mathbb{E}\Big[\sum_{\tau_\ell \geq 1} n(\tau_\ell)\mathbb{1}\big(explore(\tau_\ell)\big)\mathbb{1}\big(N_{explore}(T) > Th'(T)\big)\big|\cap_t \mathcal{G}_t\Big] + \sum_{t=1}^{T} 2NK/t^2$$

$$\overset{(v)}{\leq} Th'(T) + T\mathbb{P}\Big[N_{explore}(T) > Th'(T)\big|\cap_t \mathcal{G}_t\Big] + NK\pi^2/3$$

$$\overset{(vi)}{\leq} Th'(T) + T\mathbb{P}\Big[\exists \text{ phase } s \text{ starting at } (t_s + 1): N_{explore}(t_s) = Th'(T) \wedge \mathbb{1}(explore(s))\Big] + NK\pi^2/3$$

$$\overset{(vii)}{=} Th'(T) + NK\pi^2/3 + T\mathbb{P}\Big[\exists \text{ phase } s' \text{ starting at } (t_{s'} + 1):$$

$$N_{explore}(t_{s'}) \geq Th'(T) - N^2 \wedge (PR_u(s'), PR_a(s')) \notin \mathcal{A}(F_u^*, F_a^*)\Big]$$

$$\overset{(viii)}{=} Th'(T) + NK\pi^2/3 + T\mathbb{P}\Big[\exists t \leq T : N_{explore}(t) \geq (Th'(T) - N^2 - 1) \wedge (P_u(t), P_a(t)) \notin \mathcal{A}(F_u^*, F_a^*)\Big]$$

$$\overset{(ix)}{\leq} Th'(T) + NK\pi^2/3$$

The inequality $(ii)$ follows as for all $t \in Phase(\tau_\ell)$ the matching $M(t) = M(\tau_\ell) \notin Stable(F_u^*, F_a^*)$ only if the $(\tau_\ell - 1)$-th phase terminated with a super-stable match and we have $success = False$. But if $(\tau_\ell - 1)$-th phase has $success = False$ then $explore(\tau_\ell)$. The inequality $(iii)$ and $(iv)$ follows identically to the centralized system. The first term in inequality $(v)$ follows by noting that conditioned on good event $N_{explore}(T) = \sum_{\tau_\ell} n(\tau_\ell)\mathbb{1}(explore(\tau_\ell))$. Hence $N_{explore}(T) \leq Th'(T)$ bounds the first term by $Th'(T)$. The second term uses the trivial bound $T$ for $\sum_{\tau_\ell} n(\tau_\ell)\mathbb{1}(explore(\tau_\ell))$. For the inequality $(vi)$ we note that $N_{explore}(T)$ can cross $Th'(T)$ only if there is phase $s$ that has $explore(s) = True$, and $N_{explore}(t)$ is equal to $Th'(T)$ at the beginning of phase $s$. For inequality $(vii)$ we note that $explore(s) = True$ only if for the $(s-1)$-th phase $(PR_u(s-1), PR_a(s-1)) \notin \mathcal{A}(F_u^*, F_a^*)$, and $N_{explore}(t)$ should at least be $(Th'(T) - N^2)$ at the beginning of phase $(s-1)$. The former holds due to the fact (1), and the latter is true as each phase lasts at most $N^2$ steps. For inequality $(viii)$ we note that the above is equivalent to an existence of a time $t$ ($t$ can be taken as the penultimate round in phase $(s-2)$) such that $N_{explore}(t) \geq (Th'(T) - N^2 - 1)$ and the recovered UCB-LCB rank $(P_u(t), P_a(t)) \notin \mathcal{A}(F_u^*, F_a^*)$. However, noting $(Th'(T) - N^2 - 1) = Th(t)$ due to Lemma E.3 we know this is not possible. This gives the final result. $\square$

## F. Proof of Regret Lower Bounds

In this section, we present proof of the theorems and lemmas related to the lower bound result. Our lower bound follows a framework similar to Combes et al. (Combes et al., 2017), which in turn relies on Graves et al. (Graves & Lai, 1997). However, Combes et al. (Combes et al., 2017) does not capture our use case as it is tailored to bandit feedback. due to the multiplicity of the stable matching set there is no easy way to encode this problem in the form of linear Semi-bandit like Combes et al. (Combes et al., 2015).

We recall our system definition. Our system is parameterized by $\theta = (\boldsymbol{\mu}, \boldsymbol{\gamma}) \in \Theta := [0, 1]^{2NK}$. We extend our notations (e.g. $\text{Stable}(F_u, F_a)$, $\mathcal{A}((\boldsymbol{\mu}, \boldsymbol{\gamma}))$) to replace both $(\boldsymbol{\mu}, \boldsymbol{\gamma})$ and $(F_u, F_a)$ with $\theta$, as $\theta$ uniquely specifies our instance. The action space is the space of all matching between bipartite graphs between two sides of size $N$ and $K$, denoted as $\text{Match}(N, K)$. In round $t$, by choosing a matching $M(t) \in \text{Match}(N, K)$ the centralized platform observes a random vector $Y(M, t) = ((Y_i(t), \tilde{Y}_j(t)) : i \in [N], j \in [K])$ where for unobserved user-arm pairs, i.e. $(i, j) \notin M$, we have deterministically $Y_i(t) = \tilde{Y}_j(t) = 0$, and for any observed user-arm pairs $Y_i(t)$ come from an i.i.d. distribution (conditioned on $(i, j) \in M(t)$) with mean $\mu_{i,j}(\theta)$. Similarly, any observed $\tilde{Y}_j(t)$ are i.i.d. with mean $\gamma_{j,i}(\theta)$. For the sake of simplicity,

for the rest of the paper we work with Bernoulli variables. Extending it to a general single-parameter distribution can be done similar to Lai et al. (Lai & Robbins, 1985).

We recall from the main paper (Equation (4)) that the set of 'bad' parameters is given as

$$\Lambda(\theta) = \{\lambda \in \Theta : \underbrace{kl(\theta, \lambda; M) = 0, \forall M \in \text{Stable}(\theta)}_{\text{for all stable match of } \theta \text{ divergence is } 0}; \underbrace{\text{Stable}(\lambda) \cap \text{Stable}(\theta) = \emptyset}_{\text{no common solution}}\}$$

$$= \{\lambda \in \Theta : \mu_{ij}(\theta) = \mu_{ij}(\lambda), \gamma_{ji}(\theta) = \gamma_{ji}(\lambda), \forall (i,j) \in \text{Locked}(\theta); \lambda \notin \cup_{(P_u, P_a) \in \mathcal{A}(\theta)} \text{FullRank}(P_u, P_a)\}.$$

Therefore, we can apply the lower bound formulation in Combes et al. (Combes et al., 2015) and arrive at the following lower bound for our system. As mentioned earlier, adapting the results of Combes et al. (Combes et al., 2015) to our multiple-solution setting requires careful consideration because we need to account for the possibility of switching between optimal stationary policies, which is allowed without cost in the original framework of Graves et al. (Graves & Lai, 1997). Crucially, Graves et al. (Graves & Lai, 1997) defines the "bad" parameter space as those parameters that do not share an optimal solution with the true parameter, $\theta$, and exhibit no divergence from $\theta$ when any of its optimal policies are applied. This definition is essential for properly adapting the theoretical framework to our context.

**Theorem F.1** (Adapted from Combes et al. (Combes et al., 2015)). *For all $\theta \in \Theta$, for any* uniformly good *policy $\pi$* $\liminf_{T \to \infty} \frac{R^\pi(\theta)}{\log(T)} \geq c(\theta)$, *where $c(\theta)$ minimizes the following optimization problem*

$$\min_{\eta(M) \geq 0} \sum_{M \in \text{Match}(N,K) \setminus \text{Stable}(\theta)} \eta(M)$$

$$\text{subject to} \sum_{(i,j)} \sum_{M:(i,j) \in M} \eta(M) kl(\theta, \lambda; (i,j)) \geq 1, \forall \lambda \in \Lambda(\theta), \tag{6}$$

*where $\Lambda(\theta)$ is defined in Equation (4).*

### F.1. Instantiation of Lower Bound for Special Cases

We now construct some regret lower bounds using Theorem F.1.

**Regret 'Free' Instances:** If for each pair $(i,j)$ there exists a stable match $\mathcal{M} \in \text{Stable}(\theta)$ then regret is $o(\log(T))$. Let for any ordered list $L$, let $rot(L) = [L_{|L|}, L_1, \ldots, L_{(|L|-1)}]$ denote one anti-clockwise rotation, $rot(L, m)$ denote applying $rot(\cdot)$ $m$ times, and $rev(L) = [L_{|L|}, L_{(|L|-1)}, \ldots, L_1]$ denote reversal of the list $L$. The instance $\{\{U_i : rot([N], i) : i \in [N]\}, \{A_j : rev(rot([N], j)) : j \in [N]\}\}$ has the stable matchings $[(i, rot([N], i)[l]) : i \in [N]]$ for $l = 0, \ldots, (N-1)$. This covers all the pairs $(i,j)$ for $i \in [N]$ and $j \in [M]$. For all these instances 0 stable regret is attainable statistically. We do not have a logarithmic regret lower bound for these instances. Any algorithm that alternates between the possible stable matching of $\theta$ for the first $\Omega(T^{1-\varepsilon})$ rounds for some $\varepsilon > 0$, and then commits to a super-stable matching of the discovered partial rank if one exists or plays any sub-linear algorithm (like ours) is a uniformly good policy. But for any instance with $\Omega(1)$ reward gaps this learns with $O(\exp(-T^{1-\epsilon}))$ error rate the true rank for all users and arms. So for $\theta$ it incurs $O(1)$ regret. Note this is not a constructive argument as we do not know a-priori which matchings are stable.

**General Serial Dictatorship with Two sided Uncertainty:** In serial dictatorship all the arms share the same preference. With out loss of generality (up to renaming users) let us assume user $i$ is preferred than user $i'$ for $i < i'$. There exists a unique stable match which again without loss of generality (up to renaming arms) we can denote as $(i,i)$ for all $i \in [N]$. Then we have dominated arm for user $i$ as $D_i(\theta) = \{j : \mu_{i,j'}(\theta) > \mu_{i,i}(\theta), j \leq i\}$ for all $i \leq N$. Dominated arms are the armed preferred by user $i$ compared to it's own stable matched arm. We can construct an instance $\lambda_{\gamma,(ij)}$ by picking a $j \in D_i(\theta)$ and then making user $i$ preferable to user $j$ for arm $j$, i.e. new $\gamma'_{j,i} = \gamma_{j,j}(\theta) + \varepsilon$. We have $kl(\theta, \lambda_{\gamma,(ij)}) = kl(\gamma_{j,i}(\theta), \gamma_{j,j}(\theta) + \varepsilon)$. Because serial dictatorship has unique stable match, we now have a blocking (user, arm) pair $(i,j)$ in $\lambda_{\gamma,(ij)}$ for the matching $\{(i,i) : i \in [N]\}$, and therefore $\lambda_{\gamma,(ij)} \in \Lambda(\theta)$. We have $\Lambda_1 = \{\lambda_{\gamma,(ij)} : i \in [N], j \in D_i(\theta)\} \subseteq \Lambda(\theta)$. Next, we consider change on the user side. Let $\lambda_{\mu,(ij)}$ be constructed by picking any $j > i$, and making new $\mu'_{i,j} = \mu_{i,i}(\theta) + \varepsilon$. This way now $(i,j)$ again create blocking pairs and as a consequence we have $\lambda_{\mu,(ij)} \in \Lambda(\theta)$. Let $\Lambda_2 = \{\lambda_{\mu,(ij)} : i \in [N], j > i\} \subseteq \Lambda(\theta)$. Then the optimization problem (6), relaxed by replacing $\Lambda(\theta)$ with $\Lambda_1 \cup \Lambda_2$ has the instantiation

$$\min_{\eta(M) \geq 0} \sum_{M \neq \{(i,i): i \in [N]\}} \eta(M)$$

$$\text{s.t.} \sum_{M \ni (i,j)} \eta(M) \geq w_{i,j}, \forall (i,j),$$

$$w_{i,j} kl(\gamma_{j,i}(\theta), \gamma_{j,j}(\theta) + \varepsilon) \geq 1, \forall i \in [N], \forall j \in D_i(\theta)$$

$$w_{i,j} kl(\mu_{i,j}(\theta), \mu_{i,i}(\theta) + \varepsilon) \geq 1, \forall i \in [N], \forall j > i$$

Because each user $i$ can match with only one arm $j$ in each matching we have a lower bound for the above optimization problem as

$$c_{u,i} = \sum_{j \in D_i(\theta)} kl(\gamma_{j,i}(\theta), \gamma_{j,j}(\theta))^{-1} + \sum_{j \in [i+1, K]} kl(\mu_{i,j}(\theta), \mu_{i,i}(\theta))^{-1}.$$

Similarly, considering the perspective of each arm $j$ we obtain a lower bound

$$c_{a,j}(\theta) = \sum_{i: j \in D_i(\theta)} kl(\gamma_{j,i}(\theta), \gamma_{j,j}(\theta))^{-1} + \sum_{i < j} kl(\mu_{i,j}(\theta), \mu_{i,i}(\theta))^{-1}.$$

Thus the final lower bound for the general serial dictatorship for binary stable regret is

$$c(\theta) \geq \max \left( \max_{i \in [N]} c_{u,i}(\theta), \max_{j \in [K]} c_{a,j}(\theta) \right).$$

This lower bound is different than the lower bound in (Sankararaman et al., 2021) in multiple ways. Firstly, our bound is for binary stable regret, instead of per user individual regret like the previous one. Further note the lower bound therein does not apply for general serial matching instance, but for further specialized OSB instances. Also the bound in the previous work only uses user sided uncertainty, not both side uncertainty. So these bounds are incomparable.

### F.2. Regret Lower Bound for General Instances

In this part, we connect the lower bound in Theorem F.1 with the admissible set gaps in $\theta$.

**Definition F.2** ($\theta$-Locked). A partial rank $(P_u, P_a)$ is $\theta$-Locked *iff* for all $i \in [N]$ two arms $j, j'$ in $J_i(\theta)$ have the order in $P_{u,i}$ identical to $F_{u,i}(\theta)$, and for all $j \in [K]$ two users $i, i'$ in $I_j(\theta)$ have the order in $P_{a,j}$ identical to $F_{a,j}(\theta)$.

For a given instance $\theta$, the boundary of the admissible set, namely $\mathcal{B}(\theta)$, is defined as the set of partial ranks $(P_u, P_a) \in \mathcal{A}(\theta)$ that is not compatible with any other $(P'_u, P'_a) \in \mathcal{A}(\theta)$.

**Lemma F.3** ($\Lambda(\theta)$ Decomposition). *Given an instance $\theta$ and the boundary admissible set $\mathcal{B}(\theta)$, for each instance $\lambda$ and the corresponding full rank $(F_u(\lambda), F_a(\lambda))$ the following are equivalent: (a) $\lambda \in \Lambda(\theta)$ and (b) $(F_u(\lambda), F_a(\lambda))$ is (i) $\theta$-Locked, and (ii) reverses at least one $\theta$-open triplet for each $(P_u, P_a) \in \mathcal{B}(\theta)$, as specified below*

- *a triplet $(i, j, j')$ such that $j \underset{P_{u,i}}{>} j'$, and either $(i, j)$ or $(i, j')$ or both not in $Locked(\theta)$,*
- *a triplet $(j, i, i')$ such that $i \underset{P_{a,j}}{>} i'$, and either $(i, j)$ or $(i', j)$ or both not in $Locked(\theta)$.*

Lemma F.3 provides a wider set of instances that have sub-logarithmic regret.

**Corollary F.4.** *If there exists a $(\underline{P_u}(\theta), \underline{P_a}(\theta)) \in \mathcal{B}(\theta)$ that contains no $\theta$-open triplet then $\Lambda(\theta) = \emptyset$. Hence, the stable regret for the instance $\theta$ is sub-logarithmic.*

Proof of Lemma F.3 and Corollary F.4 are deferred to SectionG.1.

More importantly, Lemma F.3 gives us a way to construct *critical* instances by reversing $\theta$-open triplets that form a 'cover' of the set $\mathcal{B}(\theta)$ to construct a full rank $(F_u, F_a)$ that is $\theta$-locked, and does not have any stable-match overlapping with the instance $\theta$. In fact, it is enough to consider the *minimal covers* of the set $\mathcal{B}(\theta)$, which we denote as $Cover(\theta)$.

**Definition F.5** (Cover of $\mathcal{B}(\theta)$). A set of triplets $(i, j, j')$ or $(j, i, i')$ for $i, i' \in [N]$ and $j, j' \in [K]$, $C$ is a minimal cover of $\mathcal{B}(\theta)$, i.e. $C \in Cover(\theta)$, *iff* all the following statements hold
(i) each $(P_u, P_a) \in \mathcal{B}(\theta)$ there exists a triplet $o \in C$ such that $o$ is $\theta$-open for $(P_u, P_a)$,

(ii) any proper subset $C' \subset C$, $C'$ is not a cover of $\mathcal{B}(\theta)$,

(iii) the set $\Lambda_C$ defined below is non empty

$$(\tilde{\boldsymbol{\mu}}^C, \tilde{\boldsymbol{\gamma}}^C) \in \Lambda_C \iff (\tilde{\boldsymbol{\mu}}^C, \tilde{\boldsymbol{\gamma}}^C) \text{ satisfies}$$

$$\tilde{\mu}_{i,j}^C = \mu_{i,j}(\theta), \tilde{\gamma}_{j,i}^C = \gamma_{j,i}(\theta), \quad \forall (i,j) \in Locked(\theta), \tag{7}$$

$$\tilde{\mu}_{i,j'}^C \geq \tilde{\mu}_{i,j}^C + \varepsilon, \quad \forall (i,j,j') \in C, \tag{8}$$

$$\tilde{\gamma}_{j,i'}^C \geq \tilde{\gamma}_{j,i}^C + \varepsilon, \quad \forall (j,i,i') \in C, \tag{9}$$

As it will be clear shortly, in our lower bound a group of such covers can be simultaneously used as long as these covers are not overlapping.

**Definition F.6** (Non-connected Cover Group). A set of 'set of triplets' $G = \{C_1, C_2, ..C_c\}$ is called *non-connected cover-group* iff for all $C \in G$ we have $C \in Cover(\theta)$, and for each distinct pair of covers $C, C' \in G$ there is no $(i,j) \notin Locked(\theta)$ such that

$$\{(i,j,j'), (i,j',j), (j,i,i'), (j,i',i) : i' \in [N], j' \in [K]\} \cap C \cap C' \neq \emptyset.$$

The set of all non-connected cover-groups is denoted as $\mathcal{CG}(\theta)$.

The next theorem is our main lower bound result. It first formulates the optimization problem in Theorem F.1 using the $\theta$-open triplets using dual formulation. Next, given the dual optimization in itself does not highlight the dependence clearly the theorem provides a closed form lower bound (possibly not tight). For this second part we need to define a combined $kl$ divergence per cover $C$ as

$$kl(\theta, C) \triangleq \sum_i \max_{\substack{(i,j)\notin Locked(\theta), \\ (i,j')\in Locked(\theta) \\ (i,j,j'),(i,j',j)\in C}} kl\big(\mu_{ij}(\theta), \mu_{ij'}(\theta)\big) + \sum_j \max_{\substack{(i,j)\notin Locked(\theta), \\ (i',j)\in Locked(\theta) \\ (j,i,i'),(j,i',i)\in C}} kl\big(\gamma_{ji}(\theta), \gamma_{ji'}(\theta)\big), \tag{10}$$

which we use in the regret bounds.

**Theorem F.7.** *The value of $c(\theta)$ in Theorem F.1 is lower bounded by the value $c_\varepsilon(\theta)$ that optimizes the following, for any $\varepsilon > 0$*

$$\max_{\iota(\lambda)\geq 0} \sum_{\lambda \in \cup_{C\in Cover(\theta)}\Lambda_C} \iota(\lambda), \quad s.t. \sum_{\lambda \in \cup_{C\in Cover(\theta)}\Lambda_C} \iota(\lambda)kl(\theta, \lambda; M) \leq 1, \forall M \notin Stable(\theta), \tag{11}$$

*where $\Lambda_C$ is as defined in Definition F.5.*

*Furthermore, $c(\theta)$ satisfies the following lower bound $c(\theta) \geq \max_{G\in\mathcal{CG}(\theta)} \sum_{C\in G} kl(\theta, C)^{-1}$ where, $kl(\theta, C)$ is defined in Equation (10).*

The proof is provided in Section G.

Our lower bound in the main paper, Theorem 5.2, is derived from the main lower bound in Theorem F.7. We now make some remarks on various definitions used.

**Remark on Definition F.5:** We note in the above definition the condition (i) and (ii) together defines the set of minimal vertex covers of a hyper graph over the triplets as vertices, and the partial ranks $(P_u, P_a)$, containing their respective subset of $\theta$-open triplets as edges. The condition (iii) ensures that there exists an instance $\lambda$ corresponding to a cover $C$. For example, it avoids inclusion of two triplets $(i, j, j_1)$ and $(i, j_2, j)$ in cover where $(i, j_1)$ and $(i, j_2)$ both are in $Locked(\theta)$. This is not realizable because we can not chose a $\tilde{\mu}_{i,j}^C$ that simultaneously bigger than $\mu_{i,j_2}(\theta)$ and smaller than $\mu_{i,j_1}(\theta)$, where $\mu_{i,j_1}(\theta) < \mu_{i,j_2}(\theta)$. Condition (iii) is present because we need to respect the $\theta$-locked property of any $\lambda \in \Lambda(\theta)$.

## G. Proof of Results in Section F

Before proceeding to the proof, we need to define the set $kl_{>0}(\lambda) = \{(i,j) : kl(\theta, \lambda; (i,j)) \neq 0\}$ contain the pairs where $\lambda$ changes from $\theta$, and for any set of instances $A$, $kl_{>0}(A) = \cup_{\lambda\in A} kl_{>0}(\lambda)$.

We also require the following two lemmas that help with deriving bounds for optimization problem (12).

The following lemma shows how $\Lambda(\theta)$ can be broken into non-overlapping support and can be used in bounding the optimization problem (12).

**Lemma G.1.** *Consider a group of subsets $\Lambda_l \subseteq \Lambda(\theta)$ for $l \in [L]$ for $L \geq 1$, such that for any two distinct pair, $(\Lambda, \Lambda')$, we have non-overlapping support $kl_{>0}(\Lambda') \cap kl_{>0}(\Lambda)$. Then the optimization problem (12) admits a lower bound $\sum_l (\max_{\lambda \in \Lambda_l} \max_{M \notin Stable(\theta)} kl(\theta, \lambda; M))^{-1}$.*

Next, we provide a kl divergence bound for instances under the set $\Lambda_C$ for cover $C$, as defined in F.5.

**Lemma G.2.** *For a minimal cover $C \in Cover(\theta)$ there exists a $\lambda_C$ that for all $M \notin Stable(\theta)$ satisfies $kl(\theta, \lambda_C; M) \leq kl(\theta, C)$ where $kl(\theta, C)$ is as defined in Equation (10).*

*Proof of Theorem F.7.* (Part 1) We first show that $c_\varepsilon(\theta)$, defined in this theorem statement, provides a lower bound for $c(\theta)$. The dual of the above optimization problem in in Theorem F.1 is given as

$$\max_{\iota(\lambda) \geq 0} \sum_{\lambda \in \Lambda(\theta)} \iota(\lambda), \text{ s.t. } \sum_{\lambda \in \Lambda(\theta)} \iota(\lambda) kl(\theta, \lambda; M) \leq 1, \forall M \notin Stable(\theta). \tag{12}$$

As we need to construct a lower bound, any feasible solution of the above dual optimization problem will give us a valid lower bound. In particular, for any set $\Lambda' \subseteq \Lambda(\theta)$ can be used to create a valid lower bound by setting $\iota(\lambda) = 0$ for all $\lambda \notin \Lambda'$. Therefore, Equation (11) is justified as long as $\Lambda'$ used is a subset of $\Lambda(\theta)$. For that purpose, we need to show that for each triplet $C \in Cover(\theta)$, for any $\lambda_C = (\tilde{\boldsymbol{\mu}}^{\boldsymbol{C}}, \tilde{\boldsymbol{\gamma}}^{\boldsymbol{C}})$ that satisfies Equations (7), (8), and (9) we have $\lambda_C \in \Lambda(\theta)$. We call the set of $\lambda$s satisfying these three equations for some $C \in Cover(\theta)$ as $\Lambda'(\theta)$. Then $\Lambda'(\theta) \subseteq \Lambda(\theta)$. If for a given $C \in Cover(\theta)$, $\lambda$ satisfies Equation (8) or (9), then the condition $b(ii)$ in Lemma F.3 holds. Also, if $\lambda$ satisfies Equation (7) then $\lambda$ is $\theta$-locked, and $b(i)$ in Lemma F.3 holds. Therefore, by Lemma F.3 we know $\lambda \in \Lambda(\theta)$. Hence, the solution to optimization problem in the current theorem forms a lower bound for $c(\theta)$ in Theorem F.1.

(Part 2) The next part of the proof establishes that for any non-connected cover-group $G \in \mathcal{CG}(\theta)$ we have a valid lower bound as $\sum_{C \in G} \frac{1}{kl(\theta, C)}$, where $kl(\theta, C)$ is as defined in the theorem statement.

Due to Lemma G.1 we want to create group of subsets $\Lambda_l \subseteq \Lambda(\theta)$ which are pairwise non-overlapping (same as in Lemma G.1). We choose a $G \in \mathcal{CG}(\theta)$. For each $C \in G$ we construct a singleton set $\lambda_C \subseteq \Lambda$ that satisfies the equations (7), (8), and (9) that has small $kl$ divergence as specified in Lemma G.2. The constructed $\lambda_C$ has the property $kl_{>0}(\lambda_C) \subseteq \{(i,j) : \{(i,j,\cdot),(i,\cdot,j),(j,i,\cdot),(j,\cdot,i)\} \cap C \neq \emptyset\}$. But as $G$ is a non-connecting cover-group we have for any two distinct $C, C' \in G$ as $kl_{>0}(\lambda_C) \cap kl_{>0}(\lambda_{C'}) = \emptyset$. Therefore, due to Lemma G.1 we obtain the lower bound $\sum_{C \in G} \frac{1}{kl(\theta, C)}$. Taking a max over $G \in \mathcal{CG}(\theta)$ gives us the final result. $\square$

## G.1. Additional Proofs required for Proof of Theorem F.7

*Proof of Lemma F.3.* For any $\lambda \in \Lambda(\theta)$ assume $(F_u(\lambda), F_a(\lambda))$ is not $\theta$-locked that implies for some $(i,j) \in Locked(\theta)$ we have either $\mu_{i,j}(\theta) \neq \mu_{i,j}(\lambda)$ or $\gamma_{j,i}(\theta) \neq \gamma_{j,i}(\lambda)$. Which is a contradiction. So $(F_u(\lambda), F_a(\lambda))$ must be $\theta$-locked. We next assume that the full rank $(F_u(\lambda), F_a(\lambda))$ reverses no $\theta$-open triplet for some partial rank $(\underline{P_u}(\theta), \underline{P_a}(\theta)) \in \mathcal{B}(\theta)$. Consider any $i, i' \in [N]$ and any $j, j' \in [K]$. Then $j \underset{F_{u,i}(\lambda)}{>} j'$ implies $j' \underset{P_{u,i}(\theta)}{\not>} j$. Similarly, $i \underset{F_{a,j}(\lambda)}{>} i'$ implies $i' \underset{P_{a,j}(\theta)}{\not>} i$. Hence, $(F_u(\lambda), F_a(\lambda))$ is compatible with $(\underline{P_u}(\theta), \underline{P_a}(\theta))$, and $(F_u(\lambda), F_a(\lambda)) \in FullRank(\underline{P_u}(\theta), \underline{P_a}(\theta))$. But that leads to a contradiction as $(F_u(\lambda), F_a(\lambda)) \notin \cup_{(P_u, P_a) \in \mathcal{A}(\theta)} FullRank(P_u, P_a)$. So, $(F_u(\lambda), F_a(\lambda))$ must reverse at least one $\theta$-open triplet for each $(P_u, P_a) \in \mathcal{B}(\theta)$. This implies that for all $\lambda \in \Lambda(\theta)$ the conditions $(i)$ and $(ii)$ in the lemma statement are satisfied.

To prove the other direction, for some $\lambda$ let conditions $(i)$ and $(ii)$ are satisfied. Then due to $(i)$, $\lambda$ satisfies $\mu_{ij}(\theta) = \mu_{ij}(\lambda), \gamma_{ji}(\theta) = \gamma_{ji}(\lambda), \forall (i,j) \in Locked(\theta)$. Next due to $(ii)$, $(F_u(\lambda), F_a(\lambda))$ reverses at least one $\theta$-open triplet for each $(P_u, P_a) \in \mathcal{B}(\theta)$. But that means $(F_u(\lambda), F_a(\lambda))$ is no longer compatible with any $(P_u, P_a) \in \mathcal{B}(\theta)$. This in turn implies, from the definition of $\mathcal{B}(\theta)$ that $(F_u(\lambda), F_a(\lambda))$ is not compatible with any $(P_u, P_a) \in \mathcal{A}(\theta)$. This completes the proof. $\square$

*Proof of Corollary F.4.* From Lemma F.3 it immediately follows that if there is one $(\underline{P_u}(\theta), \underline{P_a}(\theta)) \in \mathcal{B}(\theta)$ that contains no $\theta$-open triplet then $\Lambda(\theta) = \emptyset$. To conclude that we can statistically attain a sub-logarithmic lower bound (not a constructive argument) we need to show that by playing stable matching we can reach the partial rank $(\underline{P_u}(\theta), \underline{P_a}(\theta))$. To recover a partial rank $(P_u, P_a)$ we need to recover all the inequalities that form the partial rank. For $(\underline{P_u}(\theta), \underline{P_a}(\theta))$ with no $\theta$-open

triplet any inequality $i' \underset{P_{a,j}(\theta)}{\not\succ} i$ implies both $(i,j)$ and $(i,j')$ lie in $Locked(\theta)$. But, by playing the set of stable matchings in a round-robin manner for the first $\Omega(T^{1-\varepsilon})$ rounds, for some $\varepsilon > 0$ we can recover the rewards of $(i,j)$ and $(i,j')$ up to any $O(1)$ accuracy $\Theta(\exp(-T^{1-\varepsilon}))$ by round $t$. Similar conclusions follows for any inequality $i' \underset{P_{a,j}(\theta)}{\not\succ} i$. Therefore, for any instance with $O(1)$ reward gaps we can attain a $O(1)$ regret using a uniformly good policy. $\qquad\square$

*Proof of Lemma G.1.* The lower bound is obtained by setting $\sum_{\lambda \in \Lambda_l} \iota(\lambda) = (\max_{\lambda \in \Lambda_l} \max_{M \notin Stable(\theta)} kl(\theta, \lambda; M))^{-1}$, and $\iota(\lambda) = 0$ for all $\lambda \notin \cup_l \Lambda_l$. The above choice gives the stated lower bound is easy to see. We need to show the inequalities hold. We pick an arbitrary $M \notin Stable(\theta)$ and calculate the right hand side of the bound as follows:

$$\sum_{\lambda \in \Lambda(\theta)} \iota(\lambda) kl(\theta, \lambda; M) = \sum_{l \in [L]} \sum_{\lambda \in \Lambda_l} \iota(\lambda) kl(\theta, \lambda; M)$$

$$= \max_{l \in [L]} \sum_{\lambda \in \Lambda_l} \iota(\lambda) kl(\theta, \lambda; M)$$

$$\leq \max_{l \in [L]} \max_{\lambda \in \Lambda_l} kl(\theta, \lambda; M) \sum_{\lambda \in \Lambda_l} \iota(\lambda)$$

$$\leq \max_{l \in [L]} \frac{\max_{\lambda \in \Lambda_l} kl(\theta, \lambda; M)}{\max_{\lambda \in \Lambda_l} \max_{M \notin Stable(\theta)} kl(\theta, \lambda; M)}$$

$$\leq 1$$

The second equality is due to the non-overlapping support property of any pair of sub sets $(\Lambda, \Lambda')$. Due to this property for any matching $M$ there exists at most one $l \in [L]$ such that $\sum_{\lambda \in \Lambda_l} \iota(\lambda) kl(\theta, \lambda; M) = 0$. The rests are standard. $\qquad\square$

*Proof of Lemma G.2.* We fix an arbitrary cover $C \in Cover(\theta)$. First, observe that for any $C \in Cover(\theta)$ we first note that keeping $\tilde{\mu}_{i,j}^C = \mu_{i,j}(\theta)$ for all $(i,j)$ such that $(i,\cdot,j) \notin C$ and $(i,j,\cdot) \notin C$ satisfies the inequalities (8). Similarly, keeping $\tilde{\gamma}_{j,i}^C = \gamma_{j,i}(\theta)$ for all $(j,i)$ such that $(j,\cdot,i) \notin C$ and $(j,i,\cdot) \notin C$ satisfies the inequalities (9). So the changes are isolated in the triplets under $C$. Therefore, it follows that

$$kl(\theta, \lambda_C; M)$$
$$= \sum_{(i,j) \in M} kl(\mu_{i,j}(\theta), \tilde{\mu}_{i,j}^C) + kl(\gamma_{j,i}(\theta), \tilde{\gamma}_{j,i}^C)$$
$$\leq \sum_i \max_{j:(i,j) \notin Locked(\theta)} kl(\mu_{i,j}(\theta), \tilde{\mu}_{i,j}^C) + \sum_j \max_{i:(i,j) \notin Locked(\theta)} kl(\gamma_{j,i}(\theta), \tilde{\gamma}_{j,i}^C)$$
$$\leq \sum_i \max_{j:(i,j) \notin Locked(\theta)} \left( \mathbb{1}((i,\cdot,j) \in C \vee (i,j,\cdot) \in C) kl(\mu_{i,j}(\theta), \tilde{\mu}_{i,j}^C) \right)$$
$$+ \sum_j \max_{i:(i,j) \notin Locked(\theta)} \left( \mathbb{1}((j,\cdot,i) \in C \vee (j,i,\cdot) \in C) kl(\gamma_{j,i}(\theta), \tilde{\gamma}_{j,i}^C) \right)$$
$$\leq \sum_i \max_{j:(i,j) \notin Locked(\theta)} \max_{\delta = \pm \varepsilon} \max_{\substack{(i,j') \in Locked(\theta) \\ (i,j,j'),(i,j',j) \in C}} kl(\mu_{i,j}(\theta), \mu_{i,j'}(\theta) + \delta)$$
$$+ \sum_j \max_{i:(i,j) \notin Locked(\theta)} \max_{\delta = \pm \varepsilon} \max_{\substack{(i',j) \in Locked(\theta) \\ (j,i,i'),(j,i',i) \in C}} kl(\gamma_{i,j}(\theta), \gamma_{i',j}(\theta) + \delta)$$

The first inequality is true as all non-zero $kl$ distance values are isolated to $(i,j) \notin Locked(\theta)$ as per (7). The second inequality holds as the changes are further isolated to triplets inside the cover $C$. We now explain the third inequality. For a given cover $C$, by Definition F.5 there is a valid $\lambda_C$ that satisfies equations (7), (8), and (9). For such a $\lambda_C$, we claim to have for any $(i,j) \notin Locked(\theta)$

$$\min_{\substack{(i,j') \in Locked(\theta) \\ (i,j,j') \in C}} \mu_{i,j'}(\theta) - \varepsilon \leq \tilde{\mu}_{i,j}^C \leq \max_{\substack{(i,j') \in Locked(\theta) \\ (i,j',j) \in C}} \mu_{i,j'}(\theta) + \varepsilon, \tag{13}$$

$$\min_{\substack{(i,j')\in Locked(\theta) \\ (j,i,i')\in C}} \gamma_{j,i'}(\theta) - \varepsilon \leq \tilde{\gamma}_{j,i}^{C} \leq \max_{\substack{(i,j')\in Locked(\theta) \\ (j,i',i)\in C}} \gamma_{j,i'}(\theta) + \varepsilon, \tag{14}$$

Due to the inequalities (7) and (8) we have

$$\tilde{\mu}_{i,j}^{C} \leq \mu_{i,j'}(\theta) - \varepsilon, \forall j' : (i,j,j') \in C, (i,j') \in Locked(\theta),$$
$$\tilde{\mu}_{i,j}^{C} \geq \mu_{i,j'}(\theta) + \varepsilon, \forall j' : (i,j',j) \in C, (i,j') \in Locked(\theta).$$

Therefore Equation (13) holds. The inequality (14) follows in a similar manner.

It is easy to see the inequality (13) implies

$$kl(\mu_{i,j}(\theta), \tilde{\mu}_{i,j}^{C}) \leq \max_{\delta = \pm\varepsilon} \max_{\substack{(i,j')\in Locked(\theta) \\ (i,j,j'),(i,j',j)\in C}} kl(\mu_{i,j}(\theta), \mu_{i,j'}(\theta) + \delta).$$

Similarly, the inequality (14) implies

$$kl(\gamma_{j,i}(\theta), \tilde{\gamma}_{i,j}^{C}) \leq \max_{\delta = \pm\varepsilon} \max_{\substack{(i',j)\in Locked(\theta) \\ (j,i,i'),(j,i',i)\in C}} kl(\gamma_{i,j}(\theta), \gamma_{i',j}(\theta) + \delta).$$

Finally, taking infimum over $\varepsilon > 0$ we obtain the final value of $kl(\theta, C)$ in Equation 10. $\qquad\square$

