# OpenReview forum: "Competing Bandits in Matching Markets via Super Stability"
_ICML.cc/2025/Conference — ICML 2025 poster_

### Official Review · Reviewer_MWcb · 2025-03-11

**Overall Recommendation:** 3

**Summary:**

The paper identifies a problem with using the Gale-Shapely algorithm for stable matching with two-sided uncertainty: finding a weakly stable matching based on partial ranking doesn't give guarantees for the full ranking. Instead, they build on a different algorithm: finding a super stable ranking that guarantees consistency between partial and full rankings. Based on that they build an algorithm to minimize the regret and analyze the instance-dependent lower bound and an upper bound. They also propose a decentralized version of the algorithm.

**Claims And Evidence:**

Yes, mostly clear

**Essential References Not Discussed:**

None, that I see.

**Experimental Designs Or Analyses:**

Theoretical paper, does not apply.

**Methods And Evaluation Criteria:**

My main concern with the paper is the notion of optimizing regret in the case of stable matching. From the user perspective, what matters most is that the algorithm converges quickly and not so much that the intermediate arms that are chosen are low-regret. In this case, I don't see why the authors don't present a bound for a pure-exploration version of the stable matching instead of the low-regret version. This limits the impact of the paper as the main metric of interest is mismatched with the motivating problem.

**Other Comments Or Suggestions:**

Line 21: should be learn instead of elarn

**Other Strengths And Weaknesses:**

Bandits have been well studied over the past. The bar for introducing a new algorithm is high and both novelty and a good motivation for a real-world problem needs to be met. My main concern here is with the motivation of this problem and the regime in which it is proposed.

**Questions For Authors:**

It would be good to talk about the motivation for regret instead of pure-exploration. Also, what are motivating examples where N <= K? The one that I can think about is the organ donor problem potentially. Also: what is the source of uncertainty for these two-sided problems in real life? Again, connect the uncertainty to a real-world problem.

**Relation To Broader Scientific Literature:**

The users find an interesting problem with the stable matching problem and correctly connect it with the super stable matching from the Irving 1994 paper.

**Theoretical Claims:**

Mostly yes but why is N <= K? In how many cases are the number of users smaller than the number of options? For example, on of the motivations for stable matching is matching students to universities. Are the authors saying the interesting case is when there are fewer students than colleges? Fewer users than the number of crowd-sourcing tasks? If their contribution is only interesting in the case of N <= K, then that limits the impact of the paper.

---

> ### Author Rebuttal · Authors · 2025-03-30
>
> We thank the reviewer for the valuable review. Please find the responses below.
>
> **Re  motivation for regret instead of pure-exploration:**  In online learning (including matching markets), guaranteeing good cumulative performance and minimizing losses is crucial. Regret, unlike pure exploration, balances learning (exploration) with earning (exploitation) to maximize total rewards. Pure exploration focuses solely on identifying the best option, disregarding rewards gained during learning. Consequently, the importance of cumulative reward in matching markets (e.g., e-commerce marketplaces) makes regret minimization a natural objective.
>
> **Re motivation of Bandits in Matching Markets:** Bandit learning in matching markets is an active field of research with a long and rich history, starting with Das & Kamenica (2005). This paper is not introducing the problem. This area has seen significant research, with over [100 publications](https://scholar.google.com/scholar?q=%22bandit+learning%22+AND+%22matching+markets%22) since then, and a recent surge in activity beginning with Liu et al. (2020). This paper pushes the frontier of bandit learning for finding any stable matching in matching markets. It proves Extended-Gale Shapley's advantage over the standard version and leverages super-stable matching structure to establish tighter, instance-specific regret lower bounds.
>
> **Re why is $N \leq K$:** We present the results with $N \leq K$ for notational convenience, but the results and respective proofs presented in this paper can be easily generalized to $N > K$ (only notational and textual changes required).
>
> **Re motivating example for $N \leq K$:** Let us consider the two sided e-commerce marketplace, such as UpWork, where clients submit specific tasks (number of task is $N$) and there are multiple vendors that can be matched with the tasks (number of vendor is $K$). In these platforms often the number of tasks is much less compared to the number of vendors (i.e. $N \leq K$).  As mentioned above, we do not claim that $N \leq K$ is more interesting than the $N> K$ scenario. The presentation is just for notational convenience.
>
>  **Re the source of uncertainty for these two-sided problems in real life?**
> Uncertainty in two-sided matching stems from a-priori unknown payoffs between users and arms, and vice versa. In the e-commerce marketplace example, a client's reward (denoted as $\mu_{i,j}$) depends on the vendor's skillset, but the vendor's execution is randomized, varying due to latent extrinsic factors modeled by reward distributions. Similarly, a vendor's preference for a client depends on factors like payment and trustworthiness, which can also vary due to latent extrinsic factors, leading to the vendor's reward  (denoted as $\gamma_{j,i}$) being modeled by random variables. We will expand on this in the revised version of the paper.

---

### Official Review · Reviewer_nH7U · 2025-03-12

**Overall Recommendation:** 2

**Summary:**

This paper studies bandit learning in two-sided matching markets where both users and arms have unknown preferences and must learn them through bandit feedback. It introduces super-stable matching, using Irving’s (1994) concept to overcome the limitations of standard Gale-Shapley (GS) algorithms, which only guarantee weak stability under uncertainty. The proposed Extended-GS algorithm, combined with UCB-LCB-based rank estimation, enables efficient matching in a centralized setting with logarithmic pessimal stable regret. A decentralized version is developed using a 2-bit communication protocol, incurring only a constant regret increase. The paper also establishes a new instance-dependent lower bound, showing that the admissible gap is a key complexity parameter for stable matching with bandit feedback.

**Claims And Evidence:**

The efficiency of the round-robin exploration strategy is theoretically justified, but it may not be the most optimal choice in practice. There is a lack of alternative exploration strategies leaves room for stronger empirical support. Additionally, while the decentralized algorithm is claimed to be scalable with a 2-bit communication protocol, no empirical evidence is provided to demonstrate its performance in large-scale settings, where factors like network latency and asynchrony could impact its effectiveness. Finally, the claim that the method is applicable to real-world matching markets (e.g., crowdsourcing, ride-sharing, college admissions) is plausible but remains untested on real-world data—all experiments are conducted on synthetic setups, making it unclear how the approach would handle dynamic, highly imbalanced, or adversarial market conditions.

**Essential References Not Discussed:**

Missed references: Pagare & Ghosh (2024), “Explore-Then-Commit Algorithms for Decentralized Two-Sided Matching Markets” (ISIT 2024)
This paper also provides instance-dependent regret bounds but under different matching assumptions. A comparison between the admissible gap used in the current work and the regret characterizations in Pagare & Ghosh (2024) could strengthen the novelty and justification of the new lower bound in Theorem 5.2.

Dai & Jordan (2021), “Learning Stable Matches with Uncertainties in Preferences” (NeurIPS 2021). The omission of Dai & Jordan (2021) NeurIPS is a significant gap in the paper’s literature review. Including it would situate the contributions more precisely within the existing body of research on bandit learning in stable matching, particularly regarding the differences between static and dynamic uncertainty settings.

**Experimental Designs Or Analyses:**

The experimental design has notable limitations. The evaluation is entirely based on synthetic data, with no real-world datasets or application-driven case studies, making it difficult to assess how well the algorithm generalizes to real-world markets with dynamic and imbalanced preference structures. Moreover, while the centralized algorithm is tested, the decentralized version is not empirically evaluated, even though its theoretical regret bound is derived. Given that real-world matching markets often operate in decentralized settings, an empirical study on scalability, robustness, and communication constraints would significantly strengthen the paper. Additionally, the round-robin exploration strategy, while simple, is not compared against more adaptive exploration approaches, leaving open the question of whether a more intelligent exploration mechanism could further improve performance.

**Methods And Evaluation Criteria:**

The experiments are conducted only on synthetic data, with no real-world datasets or practical case studies to validate the approach’s applicability in markets like crowdsourcing, ride-sharing, or college admissions. While synthetic setups allow for controlled comparisons, real-world dynamics, preference heterogeneity, and strategic behavior could introduce additional challenges not captured in the current evaluation. Furthermore, while the decentralized algorithm’s theoretical regret bound is well-analyzed, its empirical performance is not tested—the experiments focus only on the centralized approach. Given that real-world markets often involve decentralized decision-making with communication constraints, an empirical assessment of the decentralized method’s scalability and robustness would strengthen the paper. Finally, the exploration strategy (round-robin exploration) is simple but potentially suboptimal, and it would be valuable to compare it against adaptive exploration strategies to assess trade-offs in convergence speed. Expanding the evaluation to real-world scenarios, alternative exploration techniques, and decentralized settings would enhance the credibility and impact of the proposed methods.

**Other Comments Or Suggestions:**

No.

**Other Strengths And Weaknesses:**

No.

**Questions For Authors:**

1. How tight is the instance-dependent regret lower bound in practice?

2. How does the super-stable matching framework compare to user-optimal stable regret frameworks?

The paper focuses on pessimal stable regret, whereas recent works such as Hosseini et al. (2024) and Kong & Li (2023) optimize for user-optimal stable regret.

3. Why was the decentralized algorithm not evaluated empirically?

4. How sensitive is the performance to different exploration strategies?

5. Why were no real-world datasets used, and how would the method generalize?

**Relation To Broader Scientific Literature:**

First, it connects to the bandit learning literature, particularly structured multi-armed bandits in matching markets. The problem of learning stable matchings under uncertainty was first introduced in Das & Kamenica (2005), which considered single-sided uncertainty, where only users needed to learn preferences while arms had full information. Subsequent works, such as Liu et al. (2020) and Sankararaman et al. (2021), focused on decentralized learning algorithms in matching markets but assumed single-sided uncertainty and relied on the standard Gale-Shapley (GS) algorithm, which only guarantees weak stability. The Explore-then-Gale Shapley (ETGS) framework developed in Kong & Li (2023) improved upon these results by allowing for user-optimal stable regret minimization. This paper differs from these works by introducing two-sided uncertainty and leveraging super-stable matching—a concept from Irving (1994)—to guarantee true stability rather than weak stability.

In terms of matching theory, the paper draws from classical stable matching literature, particularly the Gale-Shapley (1962) deferred acceptance algorithm, but expands upon it by incorporating super stability from Spieker (1995). Prior work in bandit-based stable matching relied on weakly stable matchings under partial information, leading to suboptimal matching assignments when full rankings were revealed. By integrating Irving’s Extended Gale-Shapley algorithm, the paper provides a more robust stability concept that ensures convergence to fully stable matchings under incomplete preferences.

Regarding regret analysis in bandit learning, the paper advances the theoretical understanding of instance-dependent regret bounds in structured matching problems. The instance-dependent regret lower bound derived in Theorem 5.2 builds upon the KL-divergence-based minimization techniques from Combes et al. (2017) and Graves & Lai (1997), but specifically tailors them to two-sided matching problems with bandit feedback.

However, while the theoretical contributions are well-situated within the literature, the paper lacks empirical comparisons to recent two-sided matching bandit algorithms (e.g., Pagare & Ghosh (2024), Zhang & Fang), particularly in scalability and real-world performance. A broader experimental evaluation across different matching frameworks could further strengthen the paper’s impact in this domain.

**Theoretical Claims:**

I checked the proofs for Proposition 2.5 & Corollary 2.7, Theorems 3.5 & 4.1, and Theorem 5.2.  The regret analysis follows standard UCB-based techniques and introduces the admissible gap, though the reliance on round-robin exploration may be loose compared to adaptive strategies. The instance-dependent lower bound is derived using Combes et al. (2017) and Graves & Lai (1997).  Some assumptions are too strong: e.g.,  the scalability of the decentralized approach assumes perfect phase synchronization, which may not hold in real-world applications, and the impact of preference ties on convergence is not explicitly addressed.

---

> ### Author Rebuttal · Authors · 2025-03-30
>
> We thank the reviewer for their helpful feedback. We first want to clarify the paper's scope.
>
> This paper advances bandit learning for finding *any* stable matching in matching markets. It demonstrates Extended-Gale Shapley's effectiveness over standard Gale Shapley and uses super-stable matching structure for instance-dependent regret lower bounds. Developing methods for real-world challenges (large-scale, communication constraints, collusion robustness) is outside this paper's scope, consistent with most work in this area, and will be highlighted as future work.
>
> Please find individual responses below.
>
> **Re Missing References:** Pagare & Ghosh (2024) is cited in the Introduction (page 1, column 2, paragraph 2) and Table 1. We could not locate Dai & Jordan (2021), “Learning Stable Matches with Uncertainties in Preferences” (NeurIPS 2021), and believe the reviewer may refer to "Learning in Multi-Stage Decentralized Matching Markets." This paper addresses preference uncertainty using nonparametric statistics and variational analysis, making it not directly related. We will add it to our citations of loosely connected prior works.
>
> **Re lack of empirical comparison:** Our algorithm is compared against the centralized two-sided version of Kong et al.'s Explore then Gale Shapley, which, as stated, is the centralized variant of Pagare & Ghosh (2024) and Zhang & Fang. Therefore, we respectfully disagree that our empirical results do not compare against the recent two-sided matching bandit algorithms. Furthermore, no non-adaptive exploration baseline exists for bandits in matching market with two-sided uncertainty (to our knowledge).
>
> **Re How tight is the instance-dependent regret lower bound in practice?** We are unclear about what is meant by `practical tightness of the lower bound`. If the reviewer asks about the upper and lower bound gap, we reiterate that a tight bound for bandits in two-sided matching markets is beyond the current state-of-the-art. Our work establishes a lower bound for binary stable regret, leaving optimal algorithm design for future work.
>
> **Re focus on pessimal stable regret:** This work explores the complexity of bandit learning for finding *any* stable matching, contrasting with finding a specific one. Our binary stable regret quantifies the hardness of this objective, for which we provide regret upper and lower bounds.  Note this was the original objective in Liu et al (2020) as well.
>
> Optimizing regret for a specific stable matching (e.g. user-, arm-, or social-optimal) is a related learning objective that can leverage our findings. In particular, using the distributive lattice structure of the stable-matching efficient exploration for this task can be designed (in future) once *any* stable matching is found.
>
> **Re Why was the decentralized algorithm not evaluated empirically?** The centralized and decentralized algorithms are closely linked. We prove the decentralized algorithm's regret is at most $(1+N^2)$ higher (uniformly across instances and time T) than the centralized one. We will include decentralized algorithm simulations in the revised version to demonstrate this empirically. Our synthetic experiments focus on highlighting the effectiveness of Extended Gale-Shapley versus Gale-Shapley variants, for which the $(1+N^2)$ regret addition provides no further insight.
>
> **Re How sensitive is the performance to different exploration strategies?** Our regret upper bounds hold (up to reward gap-independent constants) as long as the exploration strategy ensures all (user, arm) pairs are explored with a constant fraction in the long run. Round robin exploration, adopted in prior work for its simplicity, ensures uniform long-run exploration. Finding an optimal non-adaptive exploration strategy is outside this work's scope. We believe under-exploring "unlikely" (user, arm) pairs is promising, but the two-sided uncertainty and numerous stable matchings complicate non-adaptive exploration.
>
> **Re Why were no real-world datasets used, and how would the method generalize?** This work, like most prior work in this area, focuses on establishing theoretical guarantees. Evaluating with real-world datasets is typically outside this line of work's scope. The reviewer's question about generalization is unclear. Scaling to large systems is beyond this paper's scope and is often addressed by contextual reward modeling, as recently considered for one-sided matching markets in Parikh et al. "Competing Bandits in Decentralized Large Contextual Matching Markets" arXiv:2411.11794.

---

### Official Review · Reviewer_7WSg · 2025-03-13

**Overall Recommendation:** 4

**Summary:**

This paper studies the bandit learning in matching markets with two-sided unknown preferences. It investigates the structure of super stability to determine the exploration-exploitation process. Existing works mainly consider LCB-UCB methods before identifying the full ranking or using known $\Delta$ to decide the exploration budget. Exploiting super stability to adaptively determine the exploration-exploitation improves the dependence on the $\Delta$ in the regret.

The paper proposes both centralized and decentralized algorithms with stable regret upper bound guarantees. The lower bound correspondings to the newly defined $\Delta$ is also provided. Experiments validate the convergence.

**Claims And Evidence:**

Yes. The claims are supported by the theoretical analysis and emprical validations.

**Essential References Not Discussed:**

Yes. Both the table comparisons and experiments lack some necesary baselines.

In Table 1, after the UCB-D3 algorithm, the following works [1][2] studying player-optimal stable regret and the two-sided unknown setting [3] are missing.


[1]Zhang Y, Wang S, Fang Z. Matching in Multi-arm Bandit with Collision. Advances in Neural Information Processing Systems (NeurIPS), 2022, pp. 9552-9563.

[2]Kong F, Wang Z, Li S. Improved Analysis for Bandit Learning in Matching Markets. Advances in Neural Information Processing Systems (NeurIPS), 2024.

[3]Zhang, Y. and Fang, Z. Decentralized two-sided bandit learning in matching market. In The 40th Conference on Uncertainty in Artificial Intelligence.

**Experimental Designs Or Analyses:**

Yes.

Some baselines are missing. For example, the following works also consider bandit learning in matching markets with player-optimal stable regret.

Zhang Y, Wang S, Fang Z. Matching in Multi-arm Bandit with Collision. Advances in Neural Information Processing Systems (NeurIPS), 2022, pp. 9552-9563.
Kong F, Wang Z, Li S. Improved Analysis for Bandit Learning in Matching Markets. Advances in Neural Information Processing Systems (NeurIPS), 2024.

**Methods And Evaluation Criteria:**

Yes. The stable regret and binary regret (market unstability) is commonly adopted in the literature.

**Other Comments Or Suggestions:**

The logic of Corollary 2.7 is a little difficult, you can modify it to make the meaning of the sentence clearer.

**Other Strengths And Weaknesses:**

Strength:

1. Investigating the super stability is a novel and interesting idea to adaptively balance exploration and exploitation in matching markets.

2. The introduction of the admissible gap is a theoretically novel contribution that refines prior instance-dependent analyses (e.g., \Delta_min) by incorporating the structure of super-stable matchings. This parameter elegantly captures the interplay between partial rankings and true stability, advancing the literature on problem-dependent regret in matching markets.

Weaknesses:

1. Though the work improves existing dependence on $\Delta_{\min}$ using a newly defined gap. The definition relies on the "admissible partial rank set" (A(Fu,Fa)), which is abstract and heavily tied to the lattice structure of super-stable matchings. This makes a less intuitive compared to $\Delta_{\min}$ and readers may struggle to grasp its relationship to instance hardness without concrete examples. Can you provide some market example to clearly compare these two gaps?

**Questions For Authors:**

In line 12 of algorithm 1, if each player chooses the arm with the largest UCB estimation value, won't there be a conflict? If not, more explanations are required.

**Relation To Broader Scientific Literature:**

Yes. This paper on bandit learning in matching markets is related to the machine learning/learning theory/bandits/multi-player literature.

**Theoretical Claims:**

I do not check every detail of the proof. But the stable regret order is standard in existing exploration-then-commit algorithms.

---

> ### Author Rebuttal · Authors · 2025-03-30
>
> We thank the reviewer for the helpful comments.
>
> **Relationship between $\Delta_{\min}$ and $\Delta_{\mathcal{A}}$:**
> We first note that  for the partial rank where the top $N$ user for each arm, and the top $N$ arms for each user are separated we always have the user-optimal matching as a super stable-matching. Hence, $\Delta_{\min} \leq \Delta_{\mathcal{A}}$ holds for all the instances. For general instances, it is not possible to improve this relationship, as there are instances where they are equal.
>
> - *We now present a class of instances where $\Delta_{\min} = \varepsilon$,  and $\Delta_{\mathcal{A}} = (1 - 2\varepsilon)$ for any $\varepsilon > 0$*.
>
> Consider the situation with $N$ users and $N$ arms. Fix any $\varepsilon > 0$.  For each user $i$  let $\mu_{i,i} = (1- \varepsilon)$ and $\mu_{i,(i+1) mod\\, N} = (1- 2\varepsilon)$ (top 2 arms), and each of the remaining arms has a mean reward $\mu_{i,j} = \varepsilon$ for all $j \notin \\{ i, (i+1) mod\\, N \\}$.  For each arm $j$ let $\gamma_{j, (j - 1) mod\\, N} = \varepsilon$ and $\gamma_{j, i} = (1 - \varepsilon)$ for all $i \neq  (j - 1) mod\\, N$. We have $\Delta_{\min} = \varepsilon$ for this instance.
>
> Consider the partial ranks $P_{u,i} = \\{ i > j,  (i+1) mod\\, N > j:  \forall j \notin \\{ i, (i+1) mod\\, N \\} \\}$ for $i \in [N]$, and $P_{a,j} = \\{ i > (j-1) mod\\, N, \forall i \neq  (j - 1) mod\\, N \\}$ for $j \in [K]$. We have $\Delta_{\min}(P_u, P_a, \mu, \gamma) = (1 - 2\varepsilon)$ for this partial rank. But this $(P_u, P_a)$ has $\\{ (i,i): \forall i \in [N] \\}$ as a super-stable matching. Therefore, $\Delta_{\mathcal{A}} \geq (1 - 2\varepsilon)$.
>
> This shows that there can be *arbitrary* (ratio is unbounded) separation between $\Delta_{\mathcal{A}}$ and $\Delta_{\min}$.
>
> **Re Missing references:**  We will expand the Table 1 to include more references pointed out by reviewers, including [1], [2], and [3] mentioned in this review. We cite some prior works (including [3]) in the paper, but leave them from Table 1 due to space constraints.
>
> **Re Corollary 2.7:** Thank you for the careful review. We will simplify the double 'for any followed by for all' logic in the final version. In particular, we will replace `for all matching M` by `each super-stable matching M`.
>
> **Re line 12 Algorithm 1:** Note that the $M_{stable}$ match returned by the Extended-GS algorithm may return multiple arm candidates for an user $i$, namely $m_i$. The largest UCB is taken only over $m_i$, not over the entire set of arms $[K]$ in line 12.  For any two distinct users $i$, and $j $, and  their respective arm candidates $m_i, m_j \in M_{stable}$, we have $m_i$ and $m_j$ are non-overlapping. Therefore, it is guaranteed that  there will be no conflict. We will provide this explanation in the revised version.

---

> > ### Comment · Reviewer_7WSg · 2025-04-03
> >
> > Thanks for the detailed response. The comparison between $\Delta_{\min}$ and $\Delta_{A}$ is insightful. It is a great improvement from $\varepsilon$ to $1-2\varepsilon$. Compared with all existing works that depend on the standard $\Delta_{\min}$, I really appreciate the method investigated to help the area understand the problem-dependent regret in matching markets. I am happy to increase my score.

---

### Official Review · Reviewer_pFWb · 2025-03-14

**Overall Recommendation:** 4

**Summary:**

This paper addresses the problem of bandit learning in two-sided matching markets with two-sided reward uncertainty, where both users and arms must learn their preferences through repeated interactions. The authors propose an innovative approach using super-stability from Irving (1994) to enhance traditional Gale-Shapley (GS) algorithms.

They adapt the Extended Gale-Shapley (GS) algorithm to find super-stable matchings instead of just weakly stable ones.
Super-stable matchings are more robust under uncertainty and ensure true stability under complete preference rankings under two kind of incomplete learning models: (i) central and (ii) local. For the central model:
The algorithm integrates the Extended GS algorithm with UCB-LCB-based rank estimation. If a super-stable matching exists under the current estimates, the algorithm selects it; otherwise, it explores via a round-robin method. The decentralized algorithm extends the centralized setting with only a constant regret increase, using 2-bit communication between users and arms.

Instance-Dependent Regret Lower Bound:
A new instance-dependent lower bound for binary stable regret is derived.
This bound characterizes the fundamental hardness of the problem using the admissible gap and highlights the pivotal role of super-stable matchings in overcoming informational bottlenecks.
Decentralized Approach:
Users propose to arms, and arms accept based on their preferences.
Two shared binary flags coordinate matching and exploration, ensuring efficient learning without a central authority.

**Claims And Evidence:**

Centralized: $O(K \log(T) / \Delta_A^2)$
Decentralized: An additional $O(N^2)$ term over the centralized bound.
Lower Bound: $\Omega(K_{\text{eff}} \log(T) / \Delta_A^2)$ where $K_{\text{eff}}$ represents the effective number of competing pairs.

The proof ideas for these claims are correct. I didn’t go through all details.

**Essential References Not Discussed:**

NA

**Experimental Designs Or Analyses:**

Experiments are well-done. Probably, the authors can show how the results vary with N and K (in a single setup).

**Methods And Evaluation Criteria:**

NA

**Other Comments Or Suggestions:**

NA

**Other Strengths And Weaknesses:**

NA

**Questions For Authors:**

1. How will you make this fully-decentralised (i.e., without using binary signal mechanism)? This is remarked as a comment but never clearly outlined.

**Relation To Broader Scientific Literature:**

This will have a broad impact on the community in general since this is a very nice problem to address. It will potentially open doors for other such problems.

**Theoretical Claims:**

All claims are theoretical and look sound to me.

---

> ### Author Rebuttal · Authors · 2025-03-28
>
> We thank the reviewer for the positive review.
>
> **Re How will you make this fully-decentralised?**
> The binary flags are used to set *restart* to True and *success* to False by individual user or arm. Note that *restart* is set from the user side, and arm side only passively acts on the *restart*  signal. An user that wants to trigger a *restart* can send a <RESTART> signal to all the arms ($K$ rounds if user can communicate with only one arm, or $1$ round if it can communicate with all arms). Once an arm receives a <RESTART> signal, the arms can respond back the <RESTART> signal to all the proposing users. Within $1$ more round all the users will receive the <RESTART> signal, and this will complete the fully-decentralised setting of restart flag. The users can adopt a similar strategy for the *success* flag as well.

---

### Decision · Program_Chairs · 2025-05-01

**Decision:**

Accept (poster)

**Comment:**

This paper tackles the challenges of bandit learning within the context of two-sided matching markets, specifically addressing the complexities of two-sided reward uncertainty. In this dynamic environment where both users and arms must refine their preferences through repeated interactions, the research explores approaches to improve learning outcomes.

The regret analysis employs standard Upper Confidence Bound (UCB) techniques while introducing the concept of the admissible gap, a significant contribution to the field. However, while the method offers valuable insights, it's worth noting that its reliance on round-robin exploration may be less effective compared to more adaptive strategies that could potentially enhance learning efficiency and performance.

Despite the promising theoretical framework, the evaluation relies exclusively on synthetic data. This narrow focus raises questions about the algorithm's applicability in real-world scenarios, where markets are often characterized by dynamic and imbalanced preference structures. The lack of real-world datasets or application-driven case studies limits our ability to ascertain how well the proposed algorithm would perform in practical settings.

Nevertheless, it is important to highlight that most reviewers expressed positive feedback about the paper's contributions, reflecting a recognition of its potential value to the field. Given this encouraging response, I find myself inclined to recommend a weak acceptance, hoping that further refinements could bridge the gap between theory and real-world application.